# Eddies in motion: visualizing boundary-layer turbulence above an open boreal peatland using UAS thermal videos

Pavel Alekseychik[1,2], Gabriel Katul[3], Ilkka Korpela[4], Samuli Launiainen[1]

[1] Bioeconomy and Environment, Natural Resources Institute Finland, 00790 Helsinki, Finland

[2] Institute for Atmospheric and Earth System Research/Physics, Faculty of Science, P.O. Box 68, FI-00014 University of Helsinki, Finland

[3] Nicholas School of the Environment, and Department of Civil and Environmental Engineering, Duke University, Durham, NC, USA

[4] Department of Forest Sciences, University of Helsinki, P.O. Box 27, 00014 University of Helsinki

*Correspondence to*: Pavel Alekseychik (pavel.alekseychik@luke.fi)

**Abstract.** High resolution thermal infrared (TIR) imaging is opening up new vistas in biosphere-atmosphere heat exchange studies. The rapidly developing unmanned aerial systems (UAS), and specially designed cameras offer opportunities for TIR survey with increasingly high resolution, reduced geometric and radiometric noise and prolonged flight times. A state-of-the-art science platform is assembled using a Matrice 210 V2 drone equipped with a Zenmuse XT2 thermal camera and deployed over a pristine boreal peatland with the aim to test its performance in a heterogeneous sedge-fen ecosystem. The study utilizes the capability of the UAS platform to hover for prolonged times (about 20 min) at a height of 500 m a.g.l. whilst recording high-frame-rate (30 Hz) TIR videos of an area of ca. 430 x 340 m. A methodology is developed to derive thermal signatures of near-ground coherent turbulent structures impinging on the land surface, surface temperature spectra and heat fluxes from the retrieved videos. The size, orientation and movement of the coherent structures are computed from the surface temperature maps, and their dependency on atmospheric conditions is examined. A range of spectral and wavelet-based approaches are used to infer the properties of the dominant turbulent scene structures. A ground-based eddy-covariance system and an in situ meteorological setup are used for reference.

Keywords: atmospheric boundary layer, boreal peatland, coherent structures, drones, energy balance, infrared imaging, UAS, surface temperature, temperature power spectrum, turbulence.

## 1 Introduction

One of the long-standing problems in turbulence research, particularly turbulence in the planetary boundary layer (PBL), is the heat transfer between rough surfaces and the turbulent flow aloft. Eddies scour the surface and transport heat from the neighborhood of the roughness elements into the unobstructed flow. The precise nature of the eddies in terms of size and contact duration that effectively impinge and scour the surface to transport heat remains a formidable challenge and a subject of active research for several decades (Owen and Thompson, 1963; Adrian, 2007). With rapid advancements in thermal infrared (TIR) imaging and image processing, a new arsenal of experimental methods pave the way to progress on these issues, which motivates the present work.

This study focuses primarily on the properties of large coherent turbulent structures, or dominant eddies as termed by Taylor (1958). He was the first to draw attention to the regular features in air temperature timeseries, which Priestley (1959) later linked to the thermals generated by surface roughness and buoyancy. Air parcels residing near the ground attain buoyancy upon receipt of heat from the ground and rise up, to become replaced by cooler air parcels descending from above, in a cyclical manner. Such ascending and descending air parcels can reach the size of the entire boundary layer, i.e. hundreds to thousands of meters across (Kaimal and Businger 1970, Kaimal et al. 1976).

However, such a mechanistic view gradually evolved into an extensive theory describing the coherent structures in increasingly higher detail. The early research mostly relied on anemometer arrays and provided often conflicting evidence about the organization of PBL coherent structures. The cross-section, average non-dimensional temperature and diameter of a thermal strongly depend on stability, according to Frisch and Businger (1973). Work by Khalsa (1980) and by Wilczak and Tillman (1980) confirmed a decrease in plume length with increasing instability. In contrast, Antonia et al. (1979) found much less consistency in the stability effect. Kaimal (1974) observed plumes that travel as constant entities with the same velocity at all heights, their translation velocity being less than the mean wind speed. However, Wilczak and Tillman (1980) present translation velocities that are always greater (1.13 times, on average) than the mean wind, although there was a large scatter in plume translation velocities (tall plumes traveling faster than short ones); translation directions often deviated from the mean wind direction. Kaimal (1976) observed plumes merging and forming boundary layer-scale structures, while Webb (1977) demonstrated larger-scale coherent structures he termed "thermal walls" translating at the velocity of the mean wind, and much smaller thermals confined to the low altitudes that did not interact with the "walls". In moderately unstable conditions, observations were made of plume elongation with the length/width ratios of 4–12 (Davison 1975). Wilczak and Tillman (1980) report the typical lengths of 300 m and widths of 40 m, and length/width ratios of 5–10.

The visualization of the coherent turbulent structures underwent a long evolution. Their exact shape was first demonstrated by particle image velocimetry (PIV) done on wind tunnel measurements. Streamwise streakiness of the wall-bounded flow velocity field was first shown by Kline et al. (1967). The PIV experiments by Tomkins and Adrian (2003) and by Ganapathisubramani et al. (2003) revealed a structure consisting of low and high momentum regions elongated along the wind direction and measured 10-20 times the boundary layer depth. Kim and Adrian (1999) and Guala et al. (2006) referred to them as very large-scale motions (VLSMs), and Kim and Adrian (1999) proposed that they consist of hairpin vortex packets. As the PIV setups of the previous studies limited the horizontal domain to about 2 times the boundary layer depth, the large-scale streakiness and underlying hairpin structure of the VLSMs could only be shown to their full extent by direct numerical simulation (DNS) of wall-bounded turbulent flows (e.g. Jeong et al. 1997) and large eddy simulation (LES) of atmospheric surface layers (e.g. Fang 2015).

Estimating the size, shape, motion and time scales of such coherent structures under real PBL conditions remains a difficult task. The principal applicability of TIR to turbulence studies has been established in several field studies to date. Some of the earliest experiments (Hoyano et al. 1999, Sugawara et al. 2001, Chudnovsky et al. 2004, Meier et al. 2010) used TIR in an urban setting to determine the thermal properties of various surfaces, and to remotely estimate the components of the surface energy balance. Vogt et al. (2008) were the first to record thermal videos over a grass field, which was soon followed by Garai and Kleissl (2011) and a similar experiment to describe the temporal skin temperature variation on various urban surfaces (Christen and Vogt 2012). Those studies established the possibility to visualize and analyze the different scales of turbulent eddies. Garai and Kleissl (2011) reported that the largest of the coherent structures were apparently much bigger than the patch of ground they measured, and proposed that flying a thermal camera suspended on a balloon at a few hundred meters above ground would improve detection of larger eddies. Modern multirotor UASs are capable of efficiently performing this task.

None of the previous TIR experiments covered a surface area exceeding ca. 200 m in diameter. The largest-scale outdoor TIR surface experiment, so far, is that of Garai and Kleissl (2013), in which an area 275 x 207 m was imaged. A similar approach by Christen et al. (2012) used a camera with a very oblique view angle targeted at a complex urban environment, which, in the presence of highly irregular shape of the underlying surface, precluded any spatial study of turbulence. Inagaki et al. (2013) and Morrison et al. (2017) recorded TIR sequences at frequencies exceeding 30 Hz, but measured relatively small areas of about 15 x 3 m and 5 x 2 m, respectively. The previous studies are therefore characterized by low spatial coverage, and disadvantageous positioning of the camera resulting in large view-zenith angles (mounting close to the ground with an oblique view angle) – all of which are alleviated by the UAS approach employed in the present study.

In this work, we explore the capability of UAS thermal imagery for detecting variations in surface temperature at high spatial and temporal resolution. Generally, we aimed to demonstrate that the near-nadir thermal imagery used here can enable inquiry into the particularities of the coherent structures' evolution and movement deduced from their 2D thermal traces on the ground. The specific goals are (1) to test the applicability of UAS TIR imagery for near-surface turbulence studies and to develop the necessary methodology to correct and analyze the images; (2) to describe the time and length scales of the entire spectrum of

surface temperature that is responsive to eddy impingement, with the focus on large structures; and (3) to compare the UAS-based turbulence characteristics to those measured by ground-based sonic anemometry.

A two-day field experiment over a pristine boreal peatland in Southern Finland was conducted using the thermal/RGB camera mounted on an unmanned quadcopter. This site was selected for two pragmatic reasons: the presence of short-stature vegetation with low thermal inertia, so as to minimize the so-called 'honami' effect, and due to the available eddy-covariance

(EC) tower measurements and meteorological data.

## 2 Materials and methods

### 2.1 Thermal imaging by the UAS

The principal sensor used was a thermal-RGB camera DJI Zenmuse XT2 mounted on a DJI Matrice 210 v2 quadcopter (Fig.

1). The IR sensor of the XT2 camera is FLIR Tau 2 (FLIR Systems Inc.). The FLIR Tau 2 sensor measured in the 7.5-13.5 μm range and had a resolution of 640 x 512 pixels.The view angle of XT2 was 45° x 37° with a 13 mm lens. Thermal resolution was better than <0.05 K, and the maximum sampling frequency was 30 Hz.

Retrievals of IR videos were conducted in four flights near noon, two on 6 August and two on 28 August 2019 at the Siikaneva pristine boreal peatland complex in southern Finland. Clear sky conditions prevailed during all four flights. About 5 min

thermal camera warmup time was allowed between UAS power-on and take-off, as longer warmup is impractical considering the limited battery life. In each flight, the drone hovered above an EC tower at an altitude of 500 m, and was able to automatically maintain the position irrespective of the wind. However, the rotation about the downlooking optical axis required manual correction throughout the flight as the drone tended to slowly turn while hovering. However, this issue was of minor importance as any such rotation is efficiently corrected at the image registration step.

The first three 30-Hz TIR video retrievals lasted for about 20 min (Table 1), whereas the fourth retrieval was 10 min due to a gimbal malfunction. The longest possible hovering flight time of 20 min matches the producer's estimate of battery life including the additional 6 minutes reserved for ascent and descent. A TIR sequence of 20 minutes thus approaches the conventional 30 min averaging period for computing turbulence statistics and vertical heat fluxes from ground-based sonic anemometry. Immediately before each flight, a non-uniformity correction was performed for the thermal camera. This

correction introduces large step-changes in the measured temperature field, which are also non-uniform across the image. It was not applied again during the flight as this would have been detrimental for the detection of frame-to-frame temperature differences at high spatial and temporal resolution. To synchronize the drone thermal video with local time (and thus the EC record time), a reference signal was created at an arbitrary moment in each flight by quickly moving a 20x60 cm aluminum plate and by recording the exact time of this manipulation. The plate otherwise lied static on the ground for the duration of the

flight. The movement of the plate was easily detected in the thermal sequences recorded from a 500 m altitude. As a result, the UAS and EC datasets were synchronized at an error of less than 1 second.

During each 20 min flight, the XT2 camera recorded surface temperature at 30 Hz, producing 20 Gbytes of raw data in the FLIR file format. The sequences were therefore sub-sampled to 1 Hz as a means of reducing data set size and processing times, while preserving the relevant turbulence timescales. The surface emissivity was set to a constant of 0.98, as the actual

emissivities of the peatland surface constituents are not precisely known; however they are expected to be about 0.98 as a representative value of a moss-dominated boreal ecosystem emissivity (Antti-Jussi Kieloaho, personal communication). Before further analyses, the sequences were converted into Matlab® data arrays (.mat) using the FLIR ResearchIR® software;

Matlab® was used for further data processing. ResearchIR also performed the correction for the transmissivity in the 500 m atmospheric column between the drone and the ground using mean air temperature and relative humidity observed during the retrievals.

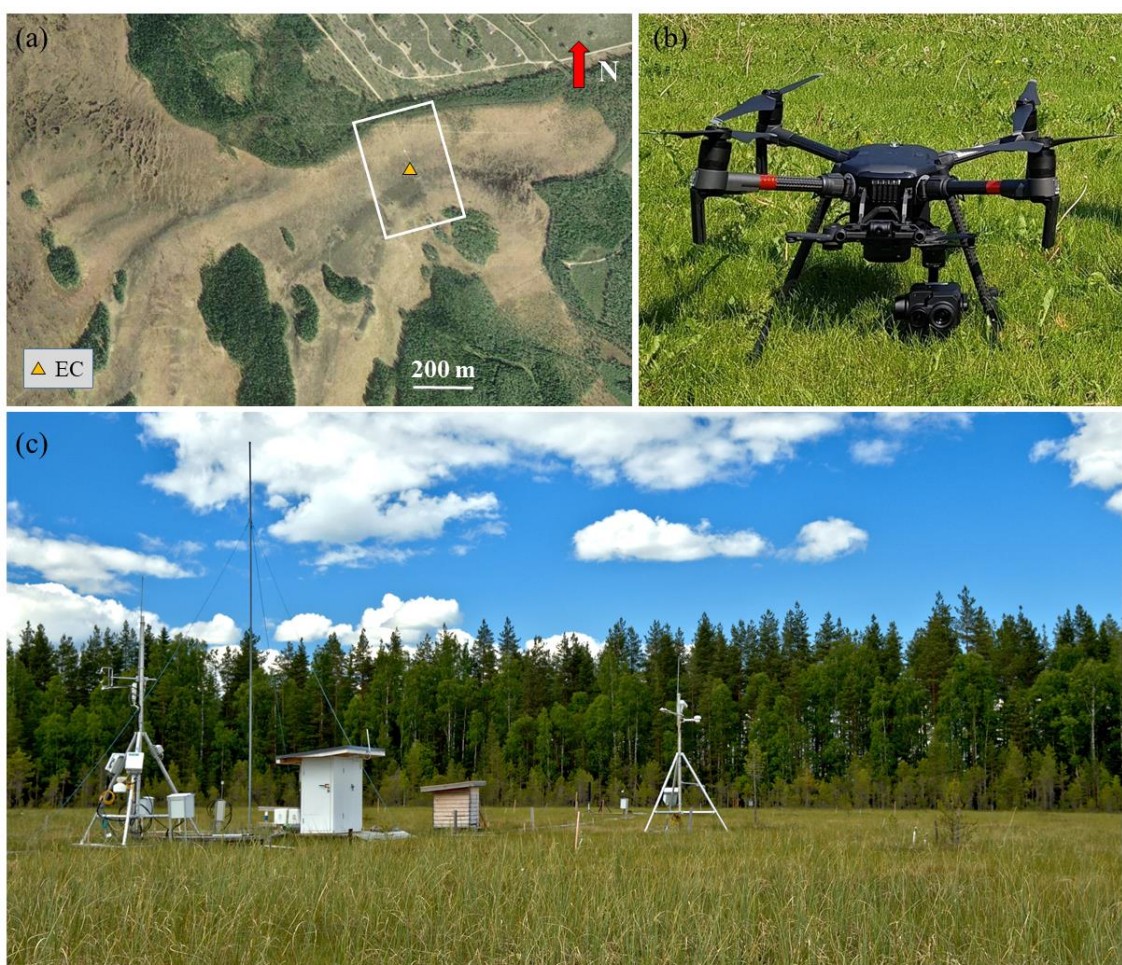

**Figure 1.** (a) Satellite view of the region surrounding the measurement site (61.832° N, 24.193°E). The area imaged by the drone is shown with the white box. A © Google Earth screenshot is used. (b) The UAS consisting of a DJI Matrice 210 v2 drone and a DJI Zenmuse XT2 thermal/RGB camera. (c) Siikaneva fen ICOS site captured at peak of sedge leaf area, viewed towards the northwest. The EC tower is on the extreme left.

**Table 1.** Flight metadata. The times specified are UTC + 3.

| Date | Altitude | Start time | End time | Total  duration | 1 Hz frame count |
|---|---|---|---|---|---|
| 06 August 2019 | 500 m | 12:28:54 | 12:47:58 | 19 min 4 s | 1144 |
| 06 August 2019 | 500 m | 13:28:35 | 13:47:17 | 18 min 42 s | 1122 |
| 28 August 2019 | 500 m | 11:22:24 | 11:42:16 | 19 min 52 s | 1193 |
| 28 August 2019 | 500 m | 12:23:09 | 12:33:36 | 10 min 27 s | 629 |

**2.2 Data post-processing**

The data post-processing workflow  in Fig. 2 consists of sequential steps from handling the raw sequences to inferring the impingement of turbulent motions from surface temperature data. Steps 1–4 can be considered to be common for all UAS-based thermal video surveys, while steps 5–6 are related to the specific aim of retrieving turbulence characteristics.

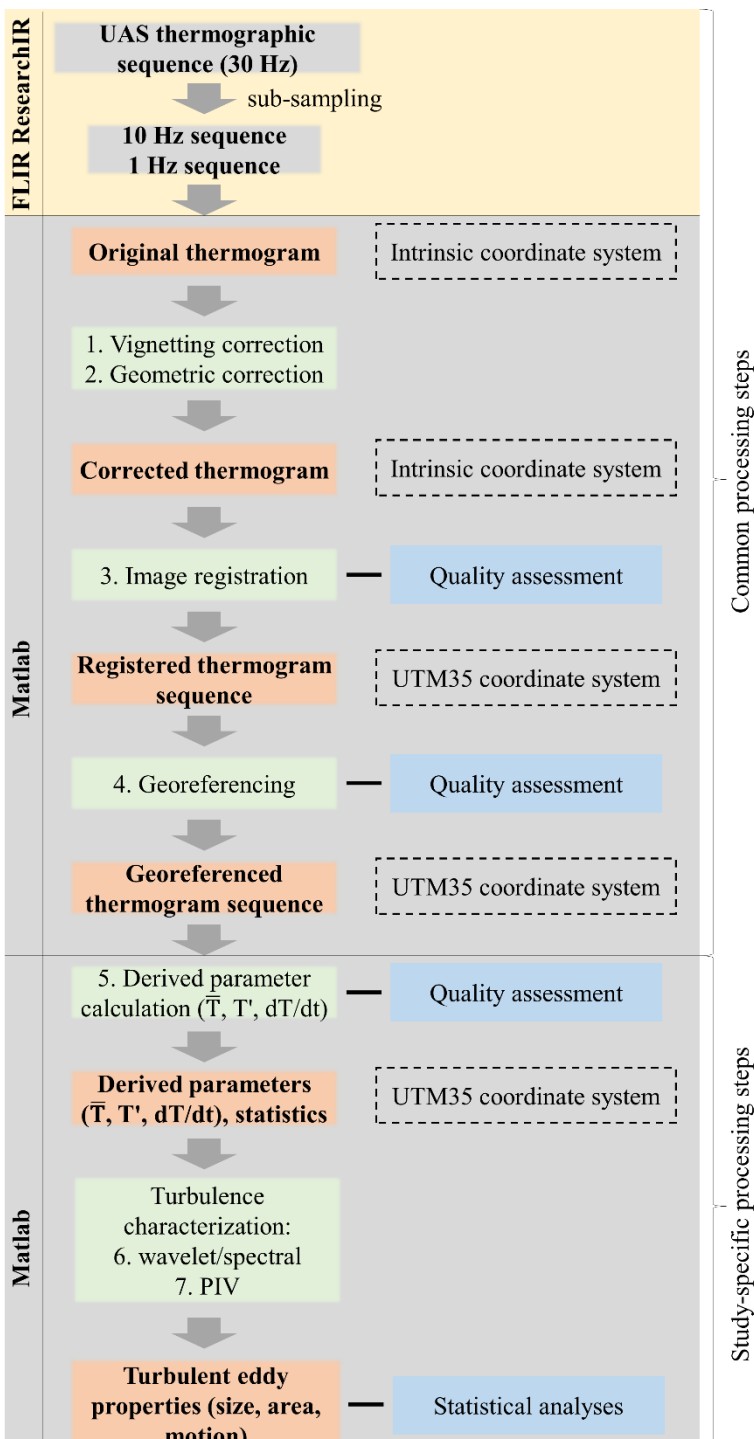

**Figure 2.** Data processing workflow. The workflow divides into the stages executed with ResearchIR and Matlab. Common processing steps are of general applicability to thermal video analysis, while the study-specific steps serve the aims of the current work. The processing steps numbered 1-7 are shown in green boxes, their output in orange boxes. Quality assessment steps and the coordinate system used at each step are shown in the right-hand column.

### 2.2.1 Vignetting correction

The vignetting effect is the artificial radial reduction in brightness temperature away from the image center, resulting from the varying lens transmissivity. A method was developed to define this lens-specific artifact and remove the effect from each image was developed. Before each flight, an image was taken of a plane fabric surface painted black to serve as a homogeneous temperature field. The reduction in temperature in the corners *versus* the center region of this thermogram (100 x 100 pixels)

was found to exceed 2° K. While the effect proved to be non-symmetrical with respect to the center of the image, it was well described by a 4th degree polynomial surface fit (Appendix A).

### 2.2.2 Geometric calibration

A practical approach was developed and implemented for geometric calibration of the TIR camera. A checkerboard similar to what is commonly used for RGB camera calibration was constructed using 5 cm paper squares glued onto an aluminum plate (see Appendix B). The aluminum plate was heated up on a stove to increase the contrast between the high-emissivity paper checkers and the low-emissivity aluminum checkers. A set of 28 images of the checkerboard taken at different angles and distances was fed to the Matlab® Camera Calibrator tool, which yielded the camera intrinsic parameters (Appendix B).

### 2.2.3 Image registration

The small, but significant, motion of the UAS during the imaging means the images needed to be co-registered, i.e. rotated and translated to a common coordinate system so that to ensure that each pixel in the output thermal video corresponds to the same point on the ground. The first frame of each of the four TIR sequences was selected as a reference, with all the subsequent frames being co-registered with it. A satisfactory solution was achieved using an intensity-based approach using *imregister* Matlab® function. Parameterization was as follows: optimization configuration = OnePlusOneEvolutionary, MattesMutualInformation = True, maximum iterations = 300, and initial radius = 0.001. With these settings, *imregister* performed an iterative solution of the image registration problem using the (1+1)-evolutionary approach using Mattes Mutual Information as a criterion of similarity between the moving and reference images. (1+1)-evolutionary optimizer involves the generation of perturbed images based on a Gaussian probability function. The perturbed image versions more similar to the template are kept, while the less similar are rejected (Styner et al. 2000). The process is continued until convergence between the perturbed and template images is achieved. Mutual Information combines the joint entropy between the images and their individual entropies as a measure of their statistical relationship; Mattes Mutual Information uses a single set of pixel locations instead of generating it at each iteration (Mattes et al. 2001).

The quality of registration was evaluated for the pairs $i^{th}$ image – reference image (the first image of the sequence) using Structural Similarity Index (SSIM), mean squared error MSE and peak signal-to-noise ratio (PSNR) as metrics (Appendix C). When the registration algorithm failed to converge (typically only a few images per flight), SSIM and PSNR displayed a downward peak and MSE an upward peak (not shown). The $i^{th}$ image that could not be registered was replaced with the $i-1^{st}$ image.

### 2.2.4 Georeferencing

The registered images were georeferenced in order to spatially relate them to the EC tower and to the georeferenced UAS RGB photo. Four ground control points (GCP) in the form of 2 x 2 m crosses with 20 cm wide arms were constructed from aluminum sheets. The GCPs formed an irregular quadrilateral with the corners at approximately 100 m distance from the EC tower. The UTM35-coordinates and ellipsoidal heights of GCPs were measured using a kinematic GNSS device (Trimble Catalyst DA1) at centimeter-level accuracy. The ground sampling density of the image pixels is 0.6 m resulting in blurred images of the crosses; the GCP was therefore determined by searching for local interpolated temperature minima within a small search area (ca. 10 m across) the expected location of the aluminum targets. Owing to large differences in emissivity, the GCP pixels were seen in strong contrast, thereby enabling georeferencing with a small sub-pixel level RMS error (Appendix D). The center of the georeferenced images has UTM latitude of 6858732 and longitude of 352185 (UTM zone 35V). The origin (0, 0) of the images corresponds to UTM latitude of 6858485 and longitude of 351968.

### 2.2.5 Derived parameters, averaging and notation

The Cartesian coordinates used are $(x, y, z)$ with x being the longitudinal (or along mean wind) direction, $y$ being lateral, and $z$ being the vertical direction with $z = 0$ being the ground surface. The three instantaneous velocity components $(u, v, w)$ are aligned along the $x, y, z$, respectively. Because the work here uses different averaging procedures including time (e.g. variables sampled at the EC tower), space, and space-time, the following conventions are used to indicate the averaging operators for an arbitrary flow variable $\chi$ evolving in space $(x, y)$ and time $(t)$. Time averaging (taken over the flight duration) at a given location $(x,y)$ is indicated by overline $\bar{\chi}$ and deviations from time averaged quantities are indicated by primes so that $\chi(x, y, t) = \bar{\chi}(x, y) + \chi'(x, y, t)$. Spatial averaging (over the sampled image domain) at a given $t$ is indicated by brackets $\langle\chi\rangle$ and deviations from this spatial average are indicated by double primes so that $\chi(x, y, t) = \langle\chi\rangle(t) + \chi''(x, y, t)$. Space-time averaging (over the image domain and flight duration, i.e. the overall mean temperature recorded during the flight) is indicated by a hat $\hat{\chi}$ and deviations from this space-time average are indicated by a tilde so that $\chi(x, y, t) = \hat{\chi} + \tilde{\chi}(x, y, t)$. For the instantaneous georeferenced surface temperature field $T(x,y,t)$, a space-time average was applied so that $\tilde{T}(x, y, t) = \hat{T} - T(x, y, t)$. A time-averaging at each pixel location was then conducted to obtain $\bar{\tilde{T}}(x, y)$ such that T'(x,y,t) $= \tilde{T}(x, y, t) - \bar{\tilde{T}}(x, y)$. Such a "zeroing" was designed to minimize the artificial changes in recorded absolute temperature due to the drift of the thermal sensor of Zenmuse XT2, FLIR Tau 2 (Dugdale et al. 2019). Temperature fluctuation distributions (see Fig. 5 and discussion therein) revealed that the physically sound $T'(x,y,t)$ values were contained in the interval $-1.5 < T' < 1.5$ K, whereas the more extreme values are deemed to represent noise; The $T'(x,y,t)$ was therefore de-spiked using those threshold bounds.

### 2.2.6 Characterizing turbulent eddy size and shape: Spectral and wavelet analysis

The post-processed $T'(x,y,t)$ are used to characterize the boundary-layer eddies impinging on the surface. Their spectral properties are first featured, followed by the transport patterns (i.e. imprint of advection velocity of large coherent eddies), size and area. A comparison between space-time surface temperature and high frequency air temperature measured at the EC location are conducted. Last, implications of $T'(x,y,t)$ to the determination of sensible heat flux from modified flux-variance similarity are discussed. Power spectra of $T'(x,y,t)$ were derived from both the drone maps and the EC-based sonic temperature. The power spectra were calculated in both the temporal and 2D spatial domains using Fast Fourier Transform.

Additionally, we used wavelet transform to infer 2D power spectra (Matlab® Wavelet Analysis Toolbox) and segregate the individual large coherent structures. Mexican Hat wavelet was applied to decompose the sequences at the spatial scales of 1-50 m; such a range was chosen based on the assumption that the maximum eddy size well represented by a TIR image covering ca. 340 x 430 m would be roughly 100 m. 2D wavelet transform was then applied to the $T'(x,y,t)$ sequences to characterize the larger coherent turbulent structures in the following manner. A 14 m scale was chosen for this particular purpose based on visual evaluation of how well the large coherent structure boundary was delineated; transform at this scale yielded isolated regions that best matched the most pronounced thermal traces. It must be noted that the wavelet transform scale is a sensitive parameter requiring adjustment to the scale of the dominant eddies; an excessively small scale value would lead to erroneous division of large eddies, while a scale value which is too high would result in grouping where the eddies are apparently separate. A 2D wavelet transform was applied to each image of the $T'(x,y,t)$ sequences, with the pixels having wavelet powers smaller than -3.5 or greater than 3.5 set to NaN to enhance the contrast between the positive and negative wavelet power regions. The threshold for this filtering operation should also be chosen with care, as the slopes separating positive and negative wavelet regions can be steep (see the effect of the ±3.5 threshold in Fig. 3c). The positive and negative regions remaining after that filtering operation represent, in essence, the smoothed boundaries of the larger coherent structure thermal traces. The wavelet transform image was discretized by setting positive regions to 1 and negative to -1, after which labeling by watershed transform. The labeled regions were then filtered by area (restricted to 500-50000 m$^2$) and mean absolute value of $T'(x,y,t)$ within their boundaries ($T'(x,y,t)$ must be >0.06 ˚K). Finally, the Matlab function *regionprops* was applied to extract the minor axis (width), major axis (length), orientation, area and the mean $T'(x,y,t)$ of each region. The regions now represent the

boundaries of large coherent structures. These operations were performed on each image of the 1 Hz $T'(x,y,t)$ sequences; Fig. 3 gives a visual example of the above operations.

Another approach to spectral analysis was taken by calculating the mean along- and cross-wind FFT spectra. To do so, each image was first rotated so as to make rows aligned with the anemometer WD averaged within ± 30 s of the image's time stamp. Then, the spectra were calculated for the rows and columns and averaged, yielding the along- and cross- wind spectra, respectively. The rows and columns containing less than 300 1-m values after rotation were omitted from the calculation. FFT was also applied to the thermal sequences in the temporal domain. FFT was first performed on individual pixel time series, and then those pixel-wise spectra were averaged to yield a single FFT spectrum of a flight.

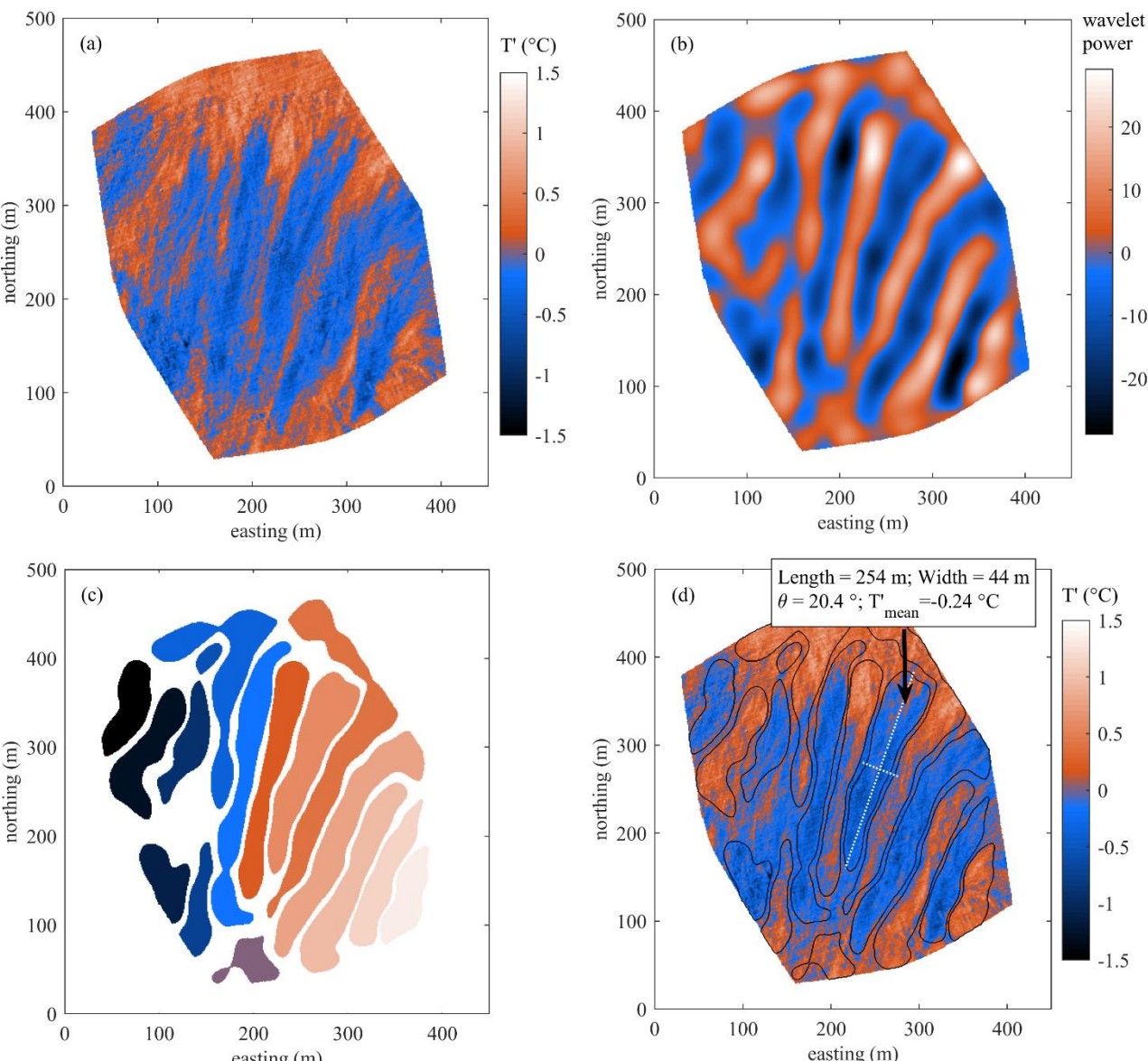

**Figure 3.** Large coherent structure identification method. (a): original georeferenced $T'(x,y,t)$ image, (b): 2D wavelet transform at 14 m scale, (c): labelled coherent structure thermal imprints obtained by watershed transform of (b), (d): original $T'(x,y,t)$ image (a) shown with the coherent structure boundaries from (c) and detailed information given for one of the identified structures (length, width, clockwise rotation from the vertical and mean temperature excursion within its boundary). The colors in (c) are only to tell the structures apart and do not correspond to the color bars in the other panels.

**2.2.7 Characterizing turbulent eddy advection velocity: Particle Image Velocimetry**

An open-source tool PIVlab (Thielicke and Stamhuis 2014) was used for thermal image velocimetry (TIV) processing to derive the speed and direction of the coherent structures motion. TIV processing was performed on the T' maps wavelet-transformed at the scale of 5 m, which provided the necessary de-noising. TIV yielded the horizontal velocity vector field of the coherent turbulent structure motions, from which the image-average advection velocity and direction were also derived. The background component of the images was removed using a built-in PIVlab GUI. After a series of tests, the following settings were chosen: interrogation area: 100 pix (i.e. 100 m at 1 m/pix); step: 50 pix, subpixel estimation method: Gauss 2x3, correlation quality: extreme, autocorrelation: disabled. As PIVlab analyzes pairs of images, this leads to a new wind field calculated for each 1 Hz image pair, i.e. once every two seconds. The output wind vectors were filtered with the threshold of 3σ in order to remove the outliers. For presentation in the case studies (Section 3.4), the TIV wind fields was averaged over a period of 80 s (see below), which provided extra smoothing. Being under 1 m, the spatial errors introduced in image registration and georeferencing do not influence the mean TIV wind fields. However, we find that the scale of wavelet transform applied to the input images does; the best performance is achieved with an evenly distributed, small and numerous "particles" (or thermal traces in the case of TIV), which motivated the choice of the 5 m wavelet decomposition scale. By trial-and-error, we established that a smaller scale leads to insufficient smoothing which distorts the TIV output, whereas a higher scale addresses the movement of large eddies, which are sparser and more difficult to process into a continuous wind field.

## 2.3 Ground-based measurements

Turbulent wind components and sonic temperature, as well as incoming global radiation and air humidity and temperature were measured on the EC tower (61°49'57.324" N, 24°11'34.116" E) of the Siikaneva fen ICOS ecosystem monitoring station. The measured ecosystem represents a treeless oligotrophic fen with a homogeneous cover of Sphagnum mosses and sedges that reach an average height of 0.25 m at the peak in July-August (Alekseychik et al. 2017a). The sonic anemometer Metek USA-1 mounted on a mast at a height of 3 m above the moss surface recorded the three velocity components ($u, v, w$) and the sonic temperature $T_s$ at a frequency of 10 Hz. The instantaneous wind speed (WS) and wind direction (WD) were calculated from these measurements, as well as the mean WS and WD during each flight. The friction velocity ($u_*$), the Obukhov length ($L_O$) and the roughness length ($z_0$) were calculated using standard equations (Stull, 1988). For the purpose of reconciling the UAS spatial thermal data with the EC record, EC flux footprints were calculated for each flight after Kormann and Meixner (2001) using 5 min averages of the 10 Hz raw EC data (for details see Alekseychik et al. 2017b).

## 3 Results

### 3.1 Micrometeorological conditions during the flights

The UAS thermographic retrievals were conducted around noon on two cloud-free August days in 2019, which proved to be rather different in terms of meteorological conditions. The first day (6 August) was characterized by substantial instability in terms of $z L_O^{-1}$ and light winds, whereas 28 August showed more near-neutral conditions and higher wind speeds (Table 2). While the stability parameter $z L_O^{-1}$ estimated from 3-m EC data pointed at near-neutrality on 28 August, the higher wind speed and friction velocity indicate a predominantly mechanical or shear-induced PBL turbulence production, as opposed to 6 August when the PBL turbulence was more buoyancy-produced. August is generally the time of the seasonal peak in sedge biomass, which causes the annual $z_0$ peak (for the investigation of $z_0$ at this site, see Alekseychik et al. 2017a); the variation in stability explains the observation of higher $z_0$ on 6 August. The kinematic sensible heat flux $\overline{w'T'_s}$ was slightly higher on 6 August. The mean wind speed and direction obtained by TIV are similar to the anemometric observations.

**Table 2.** Summary of the mean micrometeorological parameters determined at the EC station, where $\sigma_\chi$ indicates the standard deviation of an arbitrary flow variable $\chi$.

| Date, flight | Aver. period (min) | WS (ms$^{-1}$) | WD (°) | $L_o$ (m) | $u_*$ (ms$^{-1}$) | $z_0$ (m) | $\overline{w'T'_s}$ (Kms$^{-1}$) | $\sigma_w$ (ms$^{-1}$) | $\sigma_{Ts}$ (K) | $z\,L_o^{-1}$ (-) |
|---|---|---|---|---|---|---|---|---|---|---|
| 6 August, flight 1 | 19.05 | 2.15 | 63 | -10 | 0.24 | 0.15 | 0.100 | 0.34 | 0.64 | -0.29 |
| 6 August, flight 2 | 18.68 | 2.00 | 65 | -6 | 0.20 | 0.14 | 0.116 | 0.34 | 0.67 | -0.54 |
| 28 August, flight 3 | 19.87 | 3.47 | 202 | -47 | 0.39 | 0.10 | 0.092 | 0.44 | 0.49 | -0.06 |
| 28 August, flight 4 | 10.45 | 3.67 | 213 | -36 | 0.35 | 0.06 | 0.087 | 0.41 | 0.51 | -0.08 |

## 3.2 Mean temperature field $\bar{T}(x,y)$

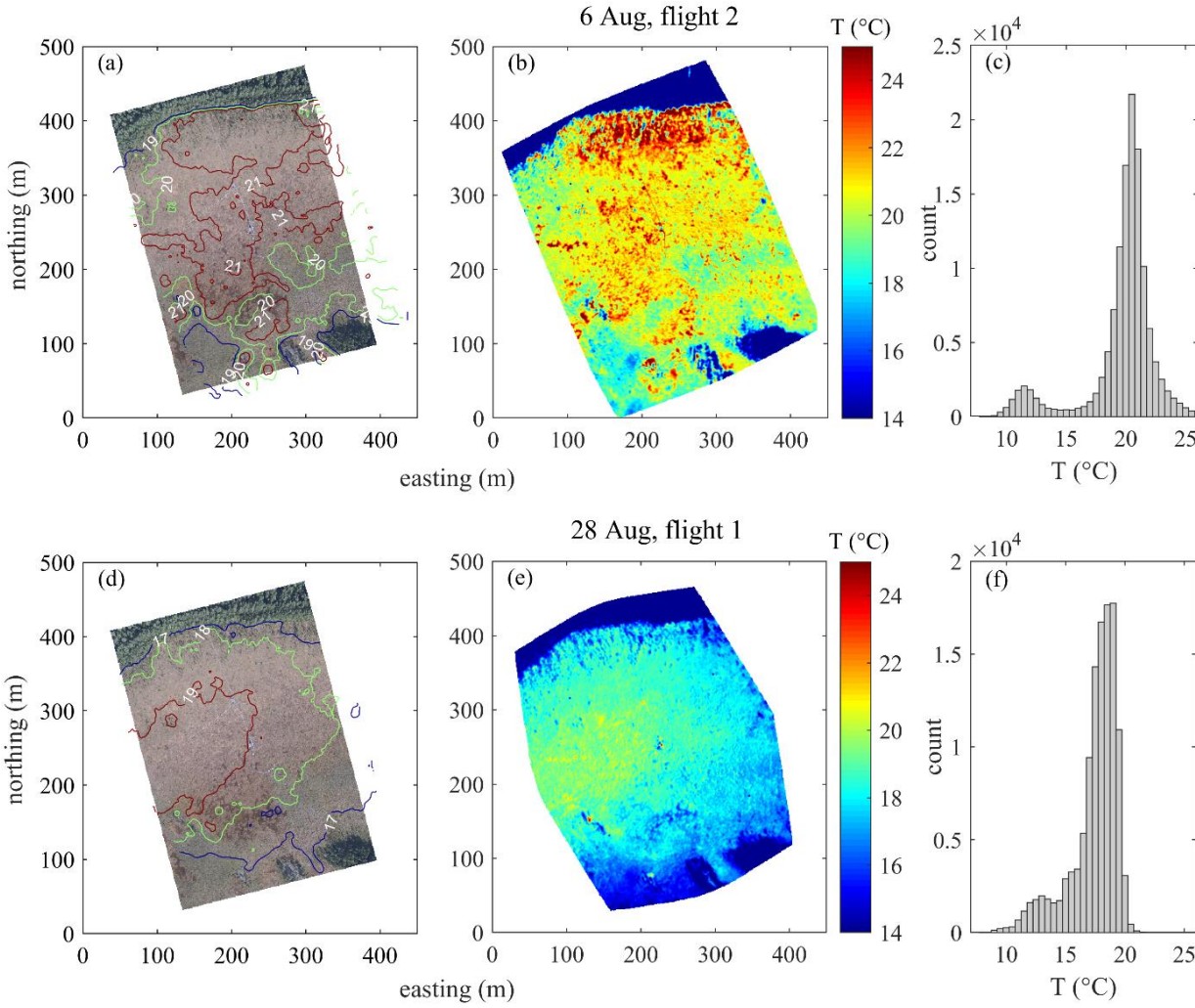

**Figure 4.** Pixel-mean surface temperature ($\bar{T}(x,y)$) measured by UAS-mounted thermal camera for the second flight of 6 August (a-c) and first flight of 28 August (d-f). The surface emissivity was assumed uniform at 0.98. (a, d) RGB photographs with temperature isolines based on the calculated $\bar{\bar{T}}(x,y)$; (b, e) surface temperature averaged for all frames of a flight ($\bar{T}(x,y)$); (c, f) histograms of $\bar{T}(x,y)$ shown in (b, e).

Fig. 4 summarizes the mean temperature variations observed on the two measurement days. Because of high similarity between $\bar{T}(x,y)$ of the flight pairs on both measurement days, only the flight 2 of 6 August and flight 1 of 28 August are shown. In terms of mean temperature, the tree stands and open peatland form two distinct regions with the tree stands appearing overall cooler and the peatland surface warmer (Fig. 4 c, f). Owing to the lack of detailed emissivity measurements, a constant emissivity of 0.98 was applied to each pixel, which might have introduced some bias in the absolute temperature values. Irrespective of the possible small biases due to the error in emissivity, the present data give a clear indication of the broad

surface temperature variations (in excess of 10 ˚C) in the open peatland on the two sampling days. The overall mean TIR temperature ($\bar{\bar{T}}(x,y)$) was higher on the 6 August (20.5 ˚C) than on 28 August (18.0 ˚C), which is evident from Fig. 4 (b, e). Secondly, $\bar{T}(x,y)$ shows different spatial distributions. While on 28 August the highest temperatures are concentrated in a circular area in the western part of field of view (FOV), on 6 August an additional zone of high $\bar{T}(x,y)$ is observed near the northern forest edge (see the isolines in Fig. 4 a, d). The peatland drainage area in the lower third of the image is characterized by the lower temperatures, part of which is formed by extensive hollow complex recognizable by its dark colour in the RGB image.

Small-scale $\bar{T}(x,y)$ variability is consistent with the hollow-hummock patterning of this ecosystem. The resulting $\bar{T}(x,y)$ patterning was intense on 6 August, with 17-20˚C in the hollows and 22-25˚C on the southern faces of the hummocks. The 28 August surface temperatures were more spatially homogeneous, probably due to stronger wind, with the hollow mean T = 18˚C and hummock mean T = 20˚C.

The observed spatial patterns in $\bar{T}(x,y)$ are not artifacts of the camera, which was assured by (i) the absence of a temporal trend in the spatial distribution of temperature maxima and (ii) the absence of significant temporal trend in the mean temperature of the frame, which, if present, would have indicated the drift due to camera stabilization and change in camera body temperature as a result of WD, WS and $T_a$ changes. That is to say, uncooled thermal camera measurements are always plagued by those artifacts, but in this case they were minimized and did not distort the environmental signal.

**3.3 Ground temperature fluctuations**

Fig. 5 features the probability density function (PDF) of $T' = \bar{T}(x,y) - <\bar{T}>$ observed in each of the four flights. The analysis suggests that the PDF is near-Gaussian with some minor deviations at the tails. A small, but significant, difference is in the kurtosis of the distributions, or, in other words, the maximum amplitude of temperature fluctuation. Allowing for the instrumental and processing-related noise, we may adopt the 2nd and 98th percentiles as estimates of the minimum and maximum T'; on the open peatland surface, those correspond to fluctuations of ± 0.7 ˚C and ± 0.6 ˚C around the mean on 6 August and 28 August, respectively. It is noteworthy that the surrounding coniferous forest always had a more fluctuating surface temperature that was about ± 1.0 ˚C and ± 0.8 ˚C on the respective days, reflecting the lower heat capacity and the higher atmospheric coupling of the conifer canopies compared to peatland surface (not shown).

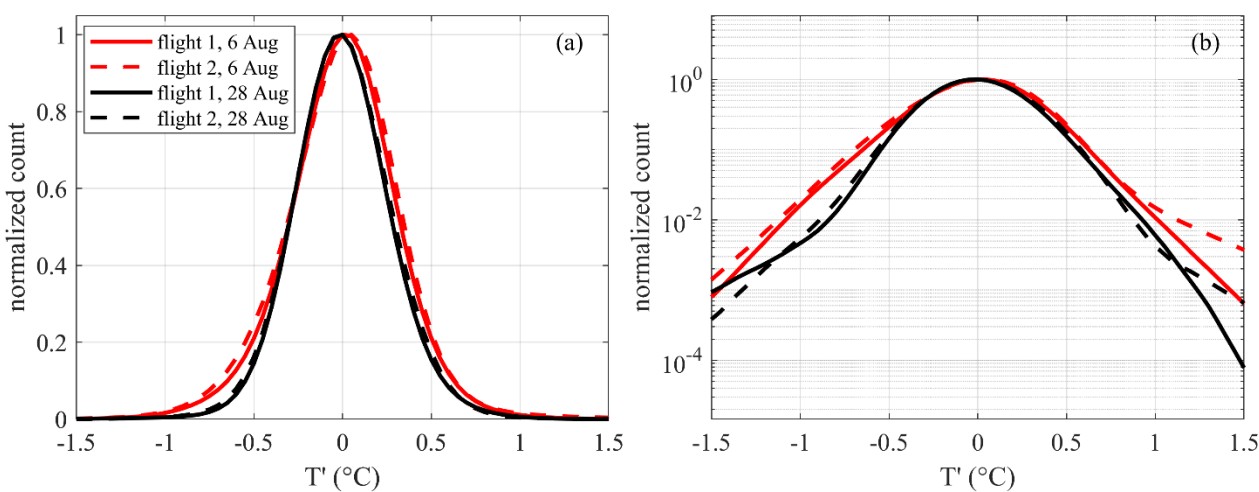

**Figure 5.** Normalized distributions of peatland surface temperature excursions during each flight, shown with linear (a) and logarithmic (b) y-axis.

The information provided in Fig. 5 is visualized spatially in Fig. 6. Not only are the excursions of ground temperature lower on the open peatland than on the other surfaces (rocky islands, tall tree stands along the south and north edges of the image),

their standard deviation is similarly contrasting. The pattern in the open peatland appeared patchier on 6 August than on 28

August. On both days, however, the standard deviation of temperature fluctuation ($\sigma_{T'}$) showed large-scale spatial inhomogeneities: it formed faint but recognizable elongated regions of alternating low and high $\sigma_{T'}$, extending along the mean wind direction.

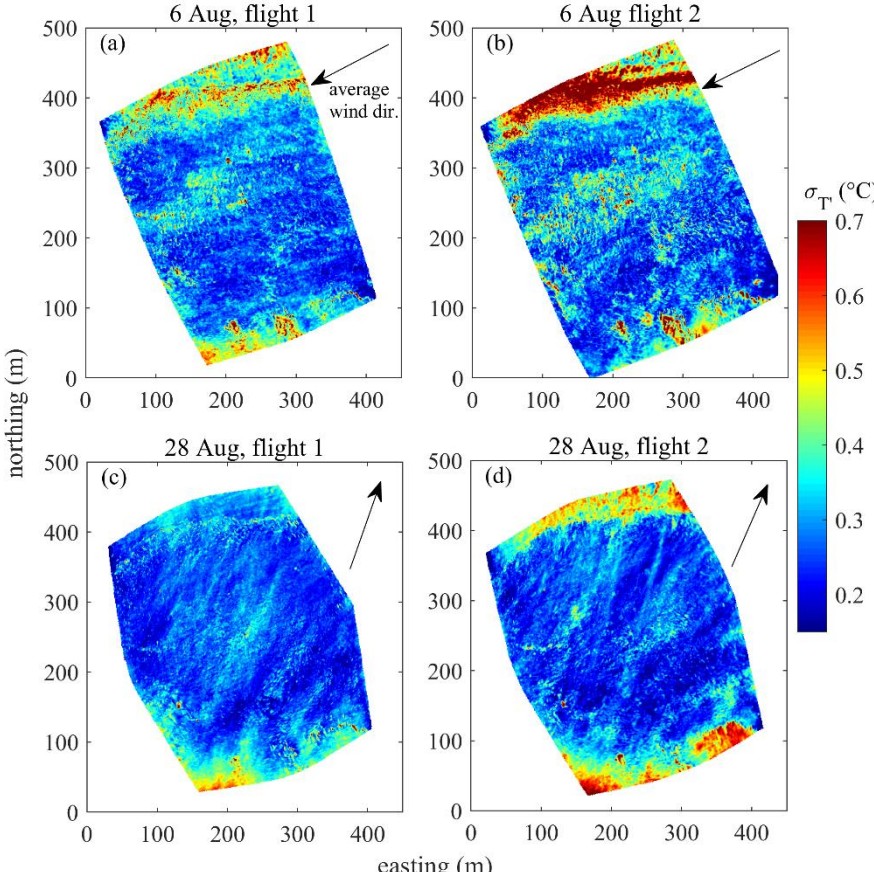

**Figure 6.** Maps of $\sigma_{T'}$. The arrows show the mean wind direction during the flight.


The average FFT spectrum of the UAS $T_g'$, and sonic anemometer-derived $T_s'$ FFT spectrum are compared in Fig. 7. The spectra are averaged over the four flights, as the differences between the flights were minor. They exhibit the same canonical -5/3 inertial subrange slope until about 0.2 Hz, where the UAS spectrum starts flattening for two primary reasons, namely the thermal camera noise and a high thermal inertia of the moist moss surface dampening the thermal influence of the small eddies.

The -1 power law relationship (Drobinsky et al. 2004, Katul et al. 1998) was not detected. The generally lower spectral energy of the UAS $T_g$ data is due to the fact that the high thermal inertia of the ground leads to much lower surface temperature fluctuations than those observed in the airflow. The flattening of the UAS spectrum at higher frequencies results from noise contributed mainly by the thermal measurement and the image registration error.

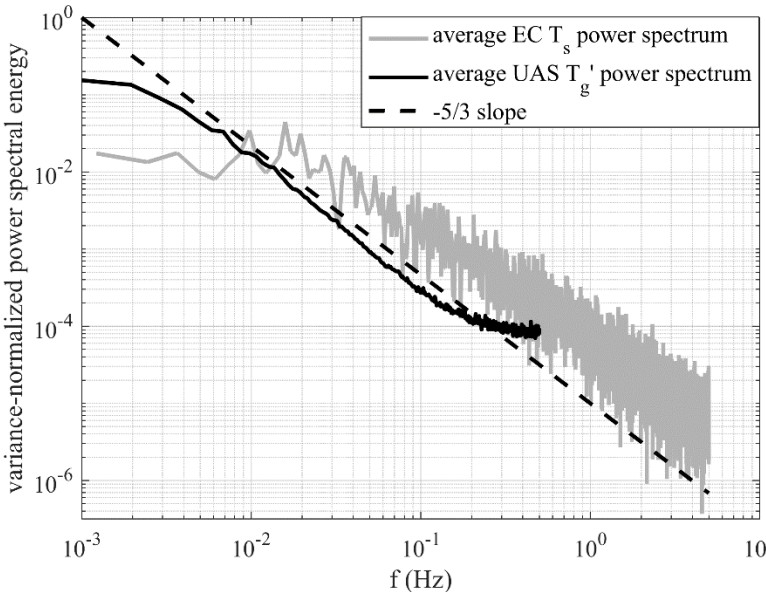

**Figure 7.** Normalized FFT power spectra of sonic temperature and drone temperature fluctuation. The UAS spectrum was calculated using 1 Hz data.

### 3.4 Detected turbulent structures and their characteristics

In order to gain a 2D aerial view of the evolving turbulent structures, the instantaneous temperature anomaly (T'(x,y,t)) maps
were combined in videos and played at a convenient rate. Distinct turbulent flow patterns were clearly visible in the T'(x,y,t) videos (see the videos in the Supplement). Based on visual inspection, three major types of large-scale flow organization or super-structure could be distinguished (Fig. 8): 1) divergent or "fanning out" pattern, 2) quiescent period when multiple convergence/divergence zones could be observed across the FOV, and 3) elongated linear structures.

Both flights of 6 August yielded evidence of flow structure cycling between the modes 1–2, with a time scale of 2–5 min. The
fanning pattern (1), when at peak strength, occupied the entire FOV and resulted in strong divergent flow pattern on the scale of several hundred meters – the wind direction is occasionally seen to differ by over 45° within a single image (Fig. 8 a). The "fan" consists of elongated eddies ca. 20–100 in length and 10–30 m in width, causing moderate ground temperature anomalies. On one occasion during the 1st flight on 6 August, the initiation of a fanning pattern is seen as an intensely cool leaf-shaped anomaly on the ground initially about 200 m in length, with the "rays" diverging from the stream-wise axis in nearly opposite
directions and rapidly emanating outwards. After this initial sweep-like stage, when the ground temperature anomaly caused by the structure reaches -1 °K, a weaker and more persistent fanning pattern as in Fig. 8 (a) persists. The intermittent quiescent periods characterized by lower wind speed and collapse of large-scale structure are demonstrated in Fig. 8 (b, c). The relatively small scale of the turbulent structures (5–50 m) probably resulted in their being confined to the roughness sublayer and thus sensitive to large roughness changes, which may explain the well-pronounced wall effects near the forest edge (Fig. 8 a, b),
which was clearly shown by the contrast in temperature standard deviation (Fig. 8 a,b,d).

A feature of particular interest is the onset of large sweeps seemingly dissociated from the mean near-surface flow, the most pronounced of which is displayed in Fig. 8 (c). The life cycle of this sweep was about 1 min from the time it reached the ground until the moment its thermal trace dissipated. Counter-flow motion of somewhat less pronounced cold structures is detectable two more times during the flights of 6 August.

Contrary to the 6 August, the large-scale flow on 28 August was completely dominated by pattern type 3, persisting throughout the flight, without the periods of true quiescence as was the case on 6 August. At a maximum, the length of those structures could exceed the largest dimension of FOV (ca. 430 m), their width being 30–100 m, but judging from visual observation most structures never reached that size. The elongation of the linear structures seemed to be positively related to the periods of

increased wind speed. Wall effects at the forest edge were virtually absent, implying significant vertical dimension of the
impinging structures (at least substantially exceeding the roughness sublayer height above the forest stand, which is about 20
m tall).

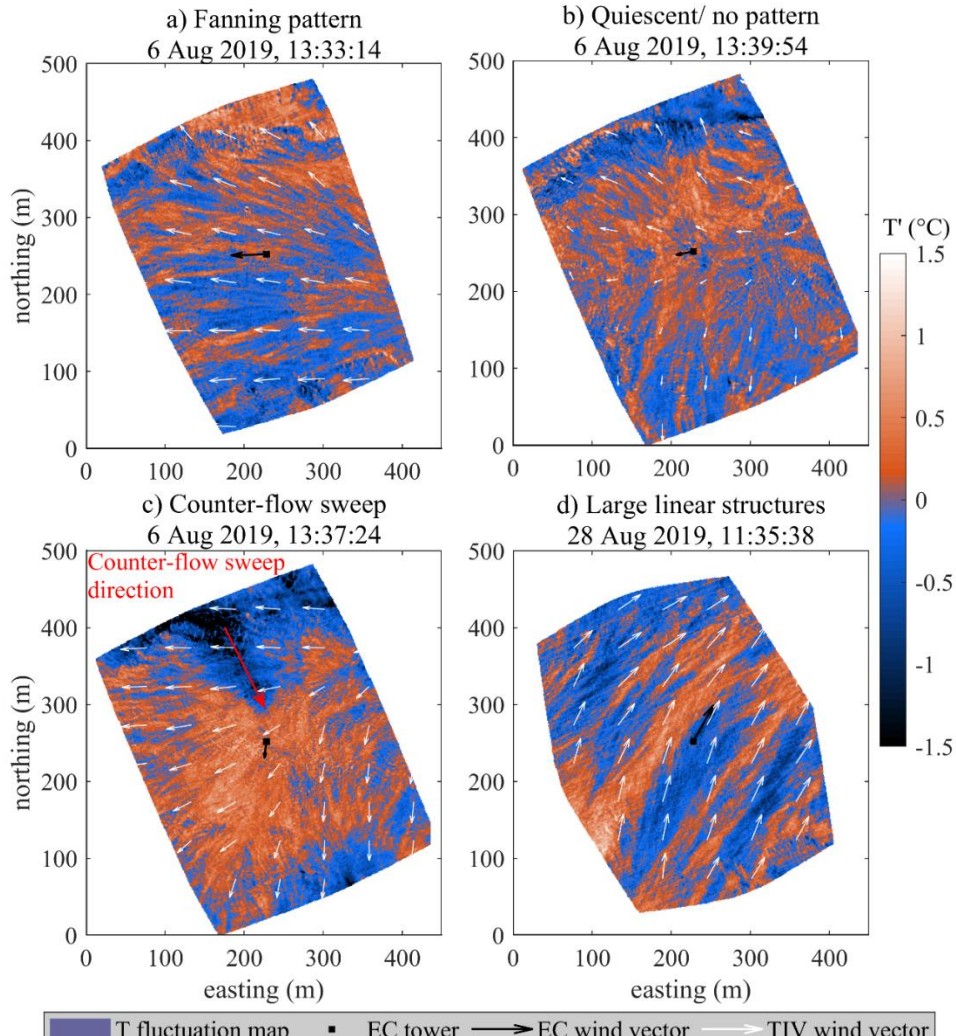

**Figure 8.** The typical cases of flow organization observed during the flights of 6 and 28 August 2019. The time stamps are
given for the presented instantaneous T' snapshots, whereas the TIV calculations were performed using the data within ±40 s
of those times. Wind vector scale is the same for each flight, TIV and EC; a twice longer vector is shown in (b) for ease of
reading.

The above case studies also demonstrate the success of TIV for boundary layer flow analysis. Table 3 reports average wind
parameters for the illustrated cases. Here, averaging over a period of 80 s centered on the image timestamp was applied so that
to both cover the interval of stationary flow required for TIV processing, and account for the typical "life time" of large
coherent structures. The average EC wind direction was within a few degrees of the average of four TIV vectors in the vicinity
of the EC tower - in the cases when the flow was stationary within FOV (Fig. 8 a, b, d). In (c), where non-stationarity appears
to be connected to the large counter-flow sweep that passed through the EC sensor, the TIV WD did not average as close to
the EC WD. The EC signal seems to have been dominated by that sweep, while the TIV flow field was unaffected by this
temporary disturbance. Another such event is seen in the very end of the second flight of 6 August (see the corresponding
video). In terms of wind speed, there is a difference between spatially homogeneous and stationary flow field (a, d) and
inhomogeneous/non-stationary flow (b, c): in the former, the EC WS was higher, in the latter, the two estimates were similar.

**Table 3.** Mean wind parameters for the cases in Fig. 8.

|  | TIV $\bar{U}$ (m s$^{-1}$) | EC $\bar{U}$ (m s$^{-1}$) | TIV WD (°) | EC WD (°) |
|---|---|---|---|---|
| Fig. 8 a | 1.8 | 2.9 | 100 | 88 |
| Fig. 8 b | 0.8 | 0.7 | 82 | 78 |
| Fig. 8 c | 1.7 | 1.4 | 41 | 6 |
| Fig. 8 d | 2.4 | 3.5 | 201 | 208 |

The periodicity of the turbulent motions was further investigated by analyzing the continuous 2D wavelet transforms on the spatial scales of 1–50 m. The result is presented in the form of scalograms normalized by the means of the spectral density at the respective scales in Fig. 9. From Fig. 9, it becomes evident that periods of intensified turbulence were more frequent on 28

August (flights 3–4) than on 6 August (flights 1–2). The normalized power generally varies more at the large spatial scales than on the small scales. A conspicuous feature is contributed by the periods of strong wavelet power increase across the larger scales, waning somewhat in the lower scales. By comparison with the T' videos, one finds that they correspond to the events of strong flow which took the form of fanning events during the flights 1 and 2. The strong fanning event in the beginning of flight 2 left a particularly sharp signature (Fig. 9 b). The waves of elongated parallel structures during the flights 3 and 4 left a

similar signature, but had shorter time spans in accordance with their lifespan. Another interesting signature also emerges, an isolated region of increased power that only encompasses a narrow range of scales. Three such "scale-dependent" bursts may be seen during flight 3, around t = [400, 650, 750] s (Fig. 9 c). At the same time, the periods of low wavelet power represent the "quiescent" conditions when well-defined large-scale structures were absent, such as that illustrated in Fig. 8b.

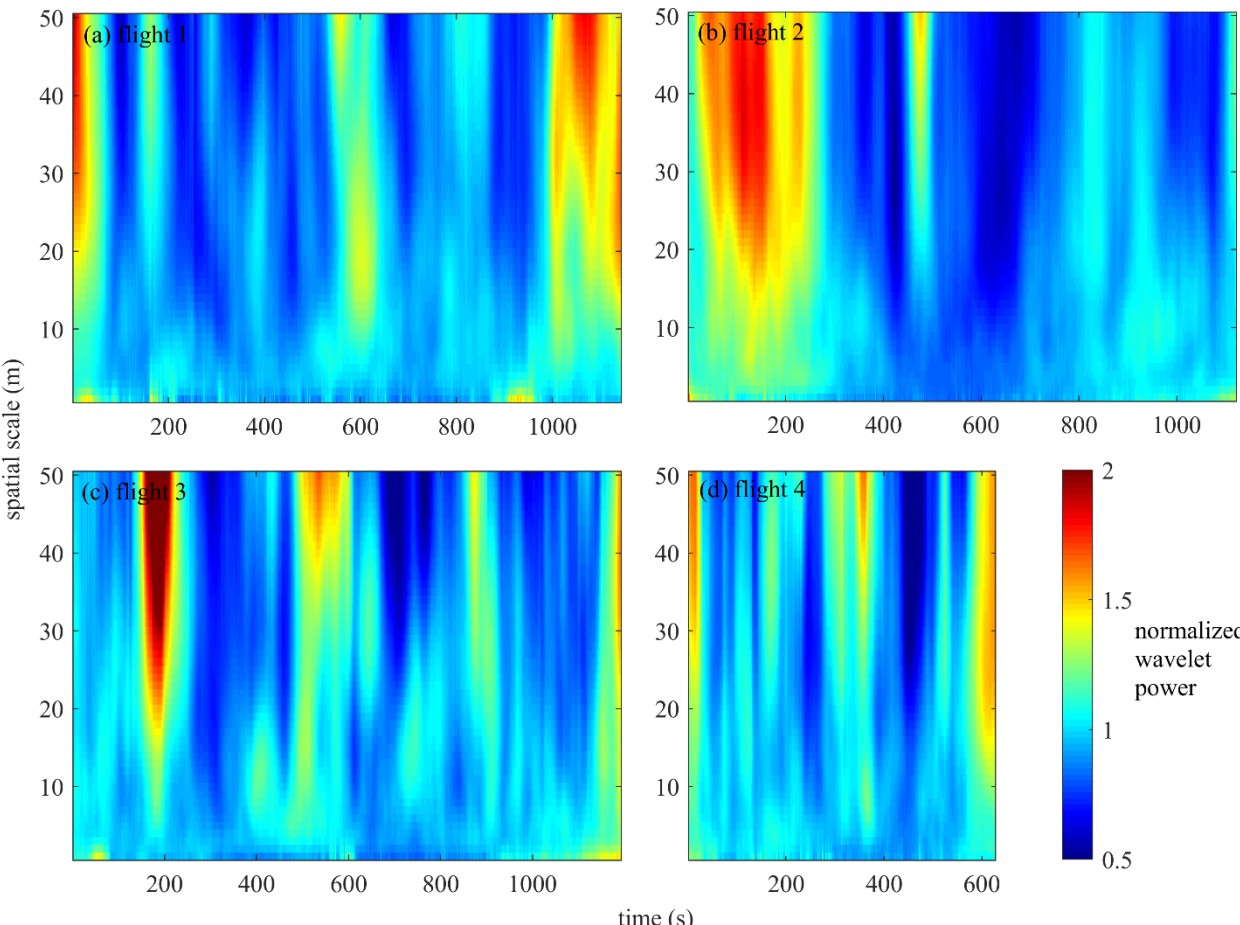


**Figure 9.** Scalograms of 2D continuous wavelet transforms on the spatial scales of 1–50 m. Absolute values of wavelet power are taken in order to average over the negative and positive T excursions, and normalized by the average absolute power at the respective scale.

The spectral properties of the ground temperature fluctuations were studied by dividing the signal into their along- and cross-wind spectra. Now, each T'(x, y, t) image was rotated and interpolated on a rectangular grid to direct the mean wind along the x-axis, y-axis being a cross-wind coordinate. The wind direction used to perform this rotation was calculated as the average anemometer WD for the period of ±30 s around the timestamp of an image. FFT power spectra were calculated for rows and columns of the rotated images on the scales of 2–128 m and averaged, yielding the mean along- and cross-wind spatial power

spectra. Rows and columns left with less than 300 pixels (= 300m) after rotation were excluded as unrepresentative of the largest spatial scales.

Two metrics are used to describe the relations between the fluctuations on different spatial scales and the along and cross wind directions, namely, (i) ratio between the spectral powers at 128 and 10 m for both along- and cross-wind directions, and (ii) ratio between the along-wind power at 128 m to cross-wind power at 128 m. (i) can be interpreted as a measure of domination

of large coherent structures at a given time, and also as a measure of anisotropy when the ratios for along- and cross-wind directions are compared. While (ii) is a measure of anisotropy, it is directly related to the largest captured scale of 128 m. The two metrics are plotted in Fig. 10. It should be noted that the 128 m-scale structures have a characteristic length scale of $\sim10^{-2}$ Hz, i.e. correspond to the energy containing subrange (Fig. 7). Consequently, the smaller eddies (under ca. 100 m in size) fall in the inertial subrange.

Most obvious is the striking dissociation between the along-wind and cross-wind ratios of the 128/10 m spectral power (metric (i)), and their intense individual variability. Generally, for both directions, the periods of increased ratios (i) correspond to the periods of intensified ground temperature fluctuations highlighted in Fig. 9, while the opposite is true for the quiescent periods as seen in Fig. 9. The tallest peaks reach the value of about 80, indicating the total dominance of large-scale coherent structures over smaller-scale near-ground turbulence. Most of the time, the ratios for the along- and cross-wind directions are anti-

correlated in the sense that typically only one of the two may peak at any given time. For example, the pronounced coherent events detected earlier at the end of flight 1, beginning of flight 2 and during flight 3 are associated with a peaking 128/10 m power ratio, indicating the contact of large coherent structures with the ground at those times. However, during flight 3 the highest values are attained by the along-wind 128/10 m power ratio, in contrast to the first two flights where the cross-wind ratio was higher.

The ratio of the 128 m spectral powers (metric ii) attains the values of 0.2–5 and displays different dynamics during the flights 1–2 and 3, being generally below unity in the former case and more often above unity in the latter case. It should be noted that the periods of increased metric (i) for cross-wind direction generally correspond to the troughs in metric (ii), as both require the cross-wind spectral power at 128 m scale to be high.

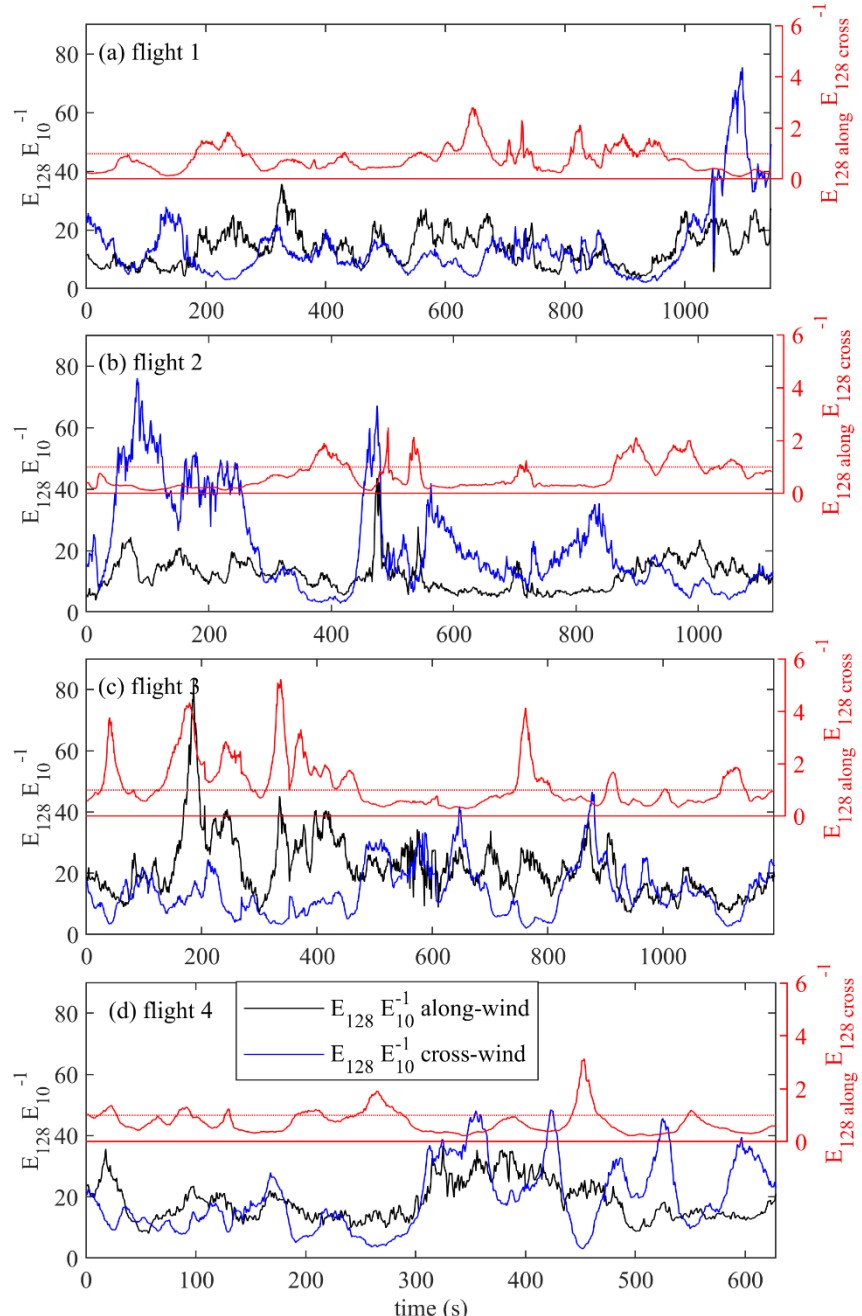


**Figure 10.** The ratio between the FFT spectral powers at the scales of 128 m to 10 m for the along- and cross-wind directions (black and blue lines, respectively) and the along- to cross- wind spectral power ratio at the scale of 128 m (red lines).

### 3.5.1 Eddy size and shape derived from 2D wavelet transforms

The eddy sizes and shapes extracted from the 2D wavelet transform are shown in Fig. 11. The algorithm (Sect. 2.2.6 and Fig. 3) detected typically 5–20 eddies per image, the eddy sizes varying between 70–240 m in length and 20–80 m in width and having areas of 1000–8000 $m^2$ (see Fig. 3 d for an example of the derived eddy parameters). The eddy parameter distributions are roughly Gaussian and display a modest but significant progression of the median eddy dimensions and area in the flight order 2–1–4–3. The flights 1–2 are similar, flight 4 being moderately different from them, whereas the flight 3 stands separate

from the former three flights. In flight 3, the detected major axes were longer while the minor axes were shorter, providing for the highest eddy length/width ratio of all flights. The eddy areas observed during that flight were generally the largest, as well. The orientation of the eddies also varied in time in close agreement with the wind direction averaged over 1 min interval centered on the thermogram record time (not shown). The thermogram-derived 14-m scale eddy orientations, however, show a systematic, variable 0–20˚ clockwise rotation relative to the anemometer wind, in each of the four flights. The elongation of

eddies therefore was collinear with the wind direction, but the directional difference increased during the quiescent periods due to the difficulty of determining the orientation of the more circular eddies which then dominated, and the wide range of eddy orientations detected within a single image.

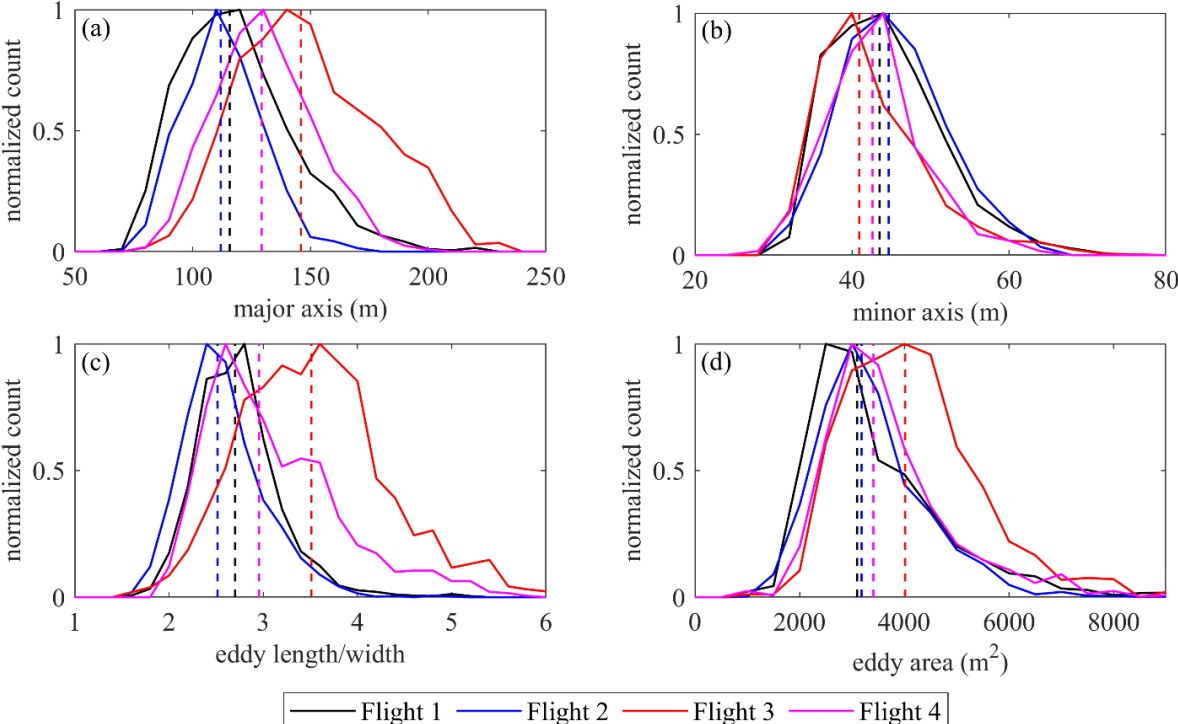

**Figure 11.** Distributions of size parameters of the eddy thermal traces derived from the 2D wavelet transforms. The vertical
dash lines mark the distribution medians. Major axis is the greater dimension of the coherent structure's thermal trace ("length"), which is always oriented in streamwise direction; correspondingly, the minor axis is the "width".

The relationship between the 5-min average eddy properties (size and shape) and diabatic stability ($z L_O^{-1}$) and $u_*$ is shown in Fig. 12 (a–d). Eddy length to width ratio is in positive correlation with both quantities (Fig. 12 a–b), implying the residence of
more elongated coherent structures during the periods of lower instability and intensified mixing. The relationship between eddy area and $z L_O^{-1}$ and $u_*$ (Fig. 11 c–d) is less strong. Eddy length/width ratio is also positively correlated with eddy area (Fig. 12 e) meaning that large eddies are typically elongated along the mean wind. The data in Fig. 12 are shown partitioned into warm and cool eddies to examine the possible differences; the offsets between the eddies of different signs reach 10–15% in relative but fail to form a clear trend against $u_*$ or stability.

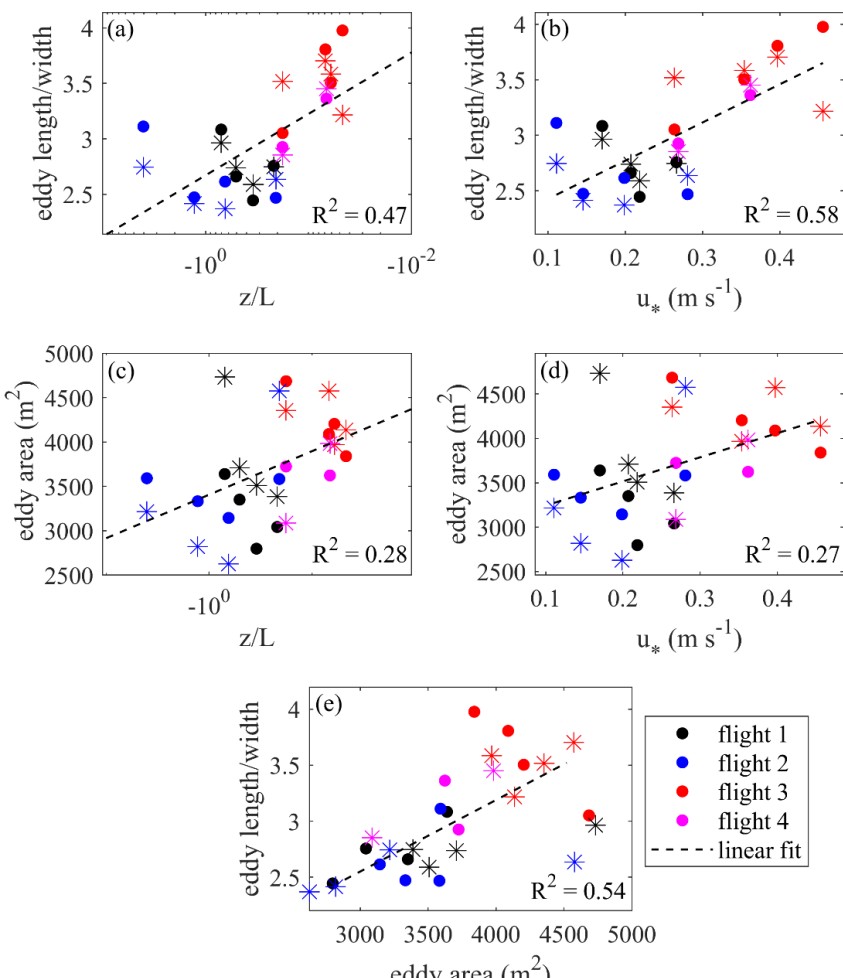

**Figure 12.** (a–d): Relation between the 5-min average properties of eddy thermal traces (mean area, length to width ratio) and the corresponding averages of $z/L_O$ and $u_*$; (e) length to width ratio *versus* mean area. The * indicate eddy traces with positive mean T', ● indicate those with negative mean T'.

### 3.5.2 Comparison of air and ground temperature excursions

The drone and EC-derived quantities ($\sigma_{Tg}$, $\sigma_{Ts}$ and mean T' difference between the positive and negative eddy regions $\Delta(T_g^-$, $T_g^+)$) were again averaged over 5-min periods to achieve a finer temporal resolution roughly matching the time scale of the coherent structures. First, we note that temperature fluctuations measured in the air and on the ground are clearly correlated (Fig. 13 a–c); the statistics of ground temperature fluctuations show $R^2$ of 0.33–0.38 against the standard deviation of sonic temperature. However, when removing 'outliers' marked with * and # in (b) and (c) the $R^2$ increases up to 0.6. Upon checking with the previously identified chronology of the coherent structure events, we find that those 5-min periods appearing to be outliers were contemporaneous with the passage of major cool coherent structures in the beginning of the flights 2 and 3 (see Fig. 9). In comparison, the regressions of the same quantities against the kinematic heat flux are considerably more scattered (Fig. 13 d–f).

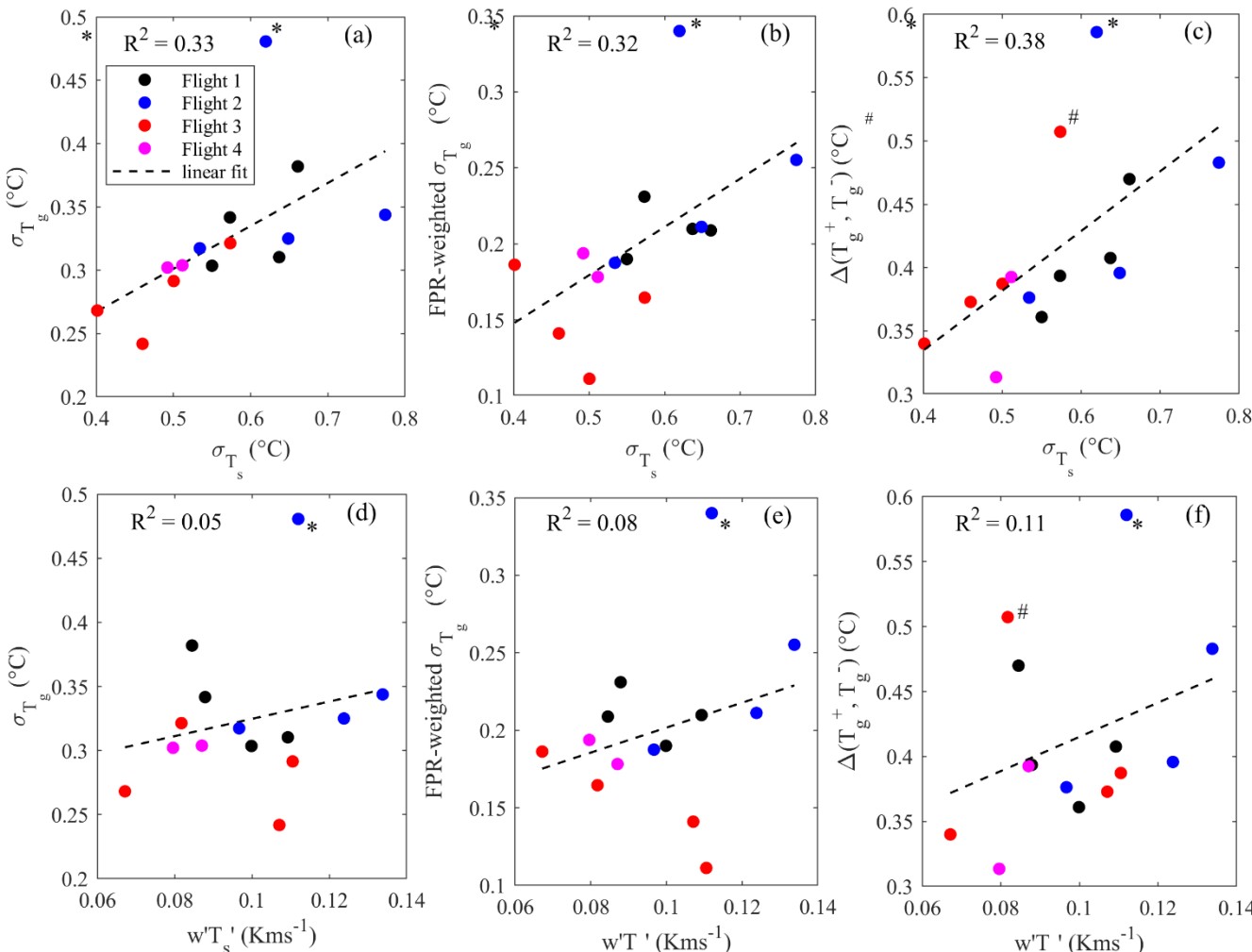

**Figure 13.** Drone data (a – standard deviation of the ground temperature, b – EC footprint -weighted standard deviation of the ground temperature (footprint based on Kormann and Meixner 2001), c – difference between the mean temperature excursions in cold and warm eddy traces) against the standard deviation of sonic temperature. In d-f, the same quantities as in a–c are plotted against the kinematic sensible heat flux. * and # indicate the outlying 5-min periods in the beginnings of the flights 2 and 3 when particularly strong coherent structures occurred.

**Discussion**

A UAS comprised of quadcopter DJI Matrice 210 V2 and camera DJI Zenmuse XT2 was capable of hovering at the altitude of 500 m for a maximum time of 20 minutes, whilst continuously recording surface temperature pointing at nadir. The setup was beneficial for the measurement of impingement of PBL turbulence on the surface: on the one hand, the record length approaches the typical averaging period of the ground-based eddy-covariance data, and was sufficient to analyze the periodicity of the large coherent turbulent structures forming in a summertime convective PBL over a peatland, on the other hand, the size of an area seen from 500 m height with a 13 mm lens was 430 x 340 m, which enables imaging coherent turbulent structures of considerable size. With these dimensions, the requirement that FOV should be bigger than the integral length scale of turbulence, proposed by Christen et al. (2012), is met. A nadir direction of view also resulted in a uniform thermal resolution at the surface of about 0.6 m/pix, which was high enough to resolve the smaller eddies down to the sizes of 1-2 m. Therefore, the present TIR study has a number of advantages over the prior work in four ways: (a) the surveyed area is the largest, (b) the camera was aimed at nadir, minimizing the geometric distortion across the thermogram; (c) the 30 Hz record rate, even when downsampled to 1 Hz, is sufficient to resolve some of the inertial-scale turbulent eddies, (d) the UAS platform can be positioned at an arbitrary point in space and thus e.g. produce imagery overlapping with the footprint of the other measurements

(such as the eddy-covariance in this case). Thus, this is the first successful attempt to use a drone to explore a wide spectrum of eddy sizes impinging on the surface via TIR imaging.

The current image analysis approach to identify turbulent coherent structures has a significant advantage over analyzing fixed sensor data on atmospheric state variables ("1D approach" in the following) in that it provides 2D images of coherent structures that are readily recognizable with naked eye, or by automated algorithms. A problem inherent to the traditional fixed-sensor meteorological observation is that the center of a coherent structure might pass at any distance from the sensor, thus making the "slice" observed in the recorded time series an unreliable representation of temperature fluctuation caused by the actual structure. In contrast, 2D imaging allows for identification of the turbulent structure shape (from its fingerprint on the ground), and its position relative to ground at any moment in time. However, even experiments employing 2D thermal imagery tended, in the past (Christen et al. 2012, Garai and Kleissl 2011, Inagaki et al. 2013), to drift towards a "statistical" perception of the coherent structures, forgoing all the potential to get a hold on the actual geometry and movement of the individual structures offered by 2D TIR. Furthermore, a 1D approach relies on turbulence being ergodic, as a result, precluding the segregation of non-stationary events from the mean flow, such as the counter-flow event in Fig. 8 (c); such non-stationarities have not been addressed in the previous studies either. This study attempted to avoid these pitfalls by focusing on spatiotemporal analyses.

The assumption underlying our approach to the imaging of turbulence is that the coherent structures dominating the surface layer flow, such as thermals, hairpins, and roll vortices remain attached to the ground for a period long enough for their evolution to be described, and this attachment is sufficiently "tight" to provide information on their internal structure. This assumption finds support in the similarity of the structures observed in this UAS experiment to those found earlier in many studies employing DNS (Fang and Porté-Agel 2015, Laima et al. 2020), Doppler radar (Newsom et al. 2008), TIV (Inagaki et al. 2013) and other methods.

The 2D organization of PBL turbulence was very disparate on the two field days. Given the difference in meteorological conditions and micrometeorological parameters, one might expect some difference in the organization of turbulence, and much evidence has emerged in this study to support this view. It appears that the conditions of 28 August, characterized by smaller instability but more intensive mixing due to stronger wind on 6 August (Table 2), led to the formation of larger and more elongated coherent structures (Figs. 8, 11). Contrastingly, when the wind picked up on the 6 August, the dominant large-scale structure was a field of smaller eddies diverging in a fan-shaped manner. The mixing on 28 August was apparently contributed by shear stress and mechanical turbulence, whereas on 6 August it was rather controlled by buoyancy and convection, which we think is the primary reason for such a substantial difference in the large-scale ABL turbulence organization. The same driver is perhaps responsible for the slower development and longer survival of self-sustaining large- scale turbulent structures on the 6 August. This result is in line with the LES study of Margairaz et al. (2020) who observed flow organization regimes depending on the magnitude of geostrophic forcing and buoyancy. There is also qualitative agreement with the proposed dependency of the size and elongation of the dominant momentum-transporting eddies on stability (Salesky et al. 2012).

The coherent structures were aligned with the wind direction, which was shown independently by comparison between the anemometric data with flow directions obtained with the help of TIV (Fig. 8, Table 3) and the orientation of eddies detected by the 2D wavelet-based algorithm (Fig. 11). However, the TIV algorithm provided for a closer match with the EC wind by virtue of its relying on the small-scale turbulence, whereas the larger-scale structures whose orientation was calculated explicitly proved to deviate by 20° clockwise, and did not follow the EC wind trend on the short timescale. Differences between the coherent structure translation direction and the EC WD were noted also in earlier works (e.g. Wilczak and Tillman 1980). The advection speed of the small-scale turbulence determined by TIV was also close to the EC wind speed.

The differences in periodicity and self-organization of turbulence on the two measurement days were assessed by spatial spectra in two ways, by 2D wavelet transform and cross- vs. along- wind FFT spectra. The 2D wavelet scalograms (Fig. 9) show transitions from high to low spectral power that are associated with impingement and dissipation of large coherent structures, which happened more frequently on the 28 August than 6 August. The two metrics constructed of cross- and along-

wind FFT spectra (Fig. 10) support the periodicity and give a general indication that the periods of increased power in Fig. 10 were the times when large structures elongated along WD dominated.

The present study paves way to determination of surface sensible heat flux based on UAS thermal videos. Firstly, there are indications that the flux variance technique (Albertson et al. 1995) may be adapted to the calculation of sensible heat flux from UAS thermal videos. As shown in Fig. 13 (a–c), the correlation between the standard deviations of sonic temperature and UAS

temperature may be sufficiently high to parameterize $\sigma_{Ts}$ as a function of $\sigma_{Tg}$ and use it in the flux variance expression of Albertson et al. (1995). The data set of the present study is, however, too short to consistently verify the validity of the approach. Care should be exercised in interpreting the UAS-derived heat fluxes, as the UAS-derived quantities which are expected to be linked to surface heat flux show a lot of scatter against EC kinematic heat flux (w'T', Fig. 13 d–f). This may be due to a variety of reasons – e.g. the invalidity of the footprint model calculated with 5 min averages or the lack of direct

link between $\sigma_{T'}$ and w'T' on a 5 min time scale. However, we propose that, if the eddies are large, attached to the ground, contribute most to heat transport, and the flow is ergodic, space-time mean $\sigma_{Tg}$ of the eddy at the ground can ultimately be used to derive heat flux caused by the impingement of an eddy. Alternatively, a 'pixel' type flux variance can be used to infer variations in sensible heat flux, where heat flux is calculated for individual pixels using their specific $\sigma_{Tg}$. Secondly, the possibility to segregate large eddies and derive the durations of their contact with the ground enables the use of a modified

surface renewal approach (Paw U et al. 1995), in which the amplitude of temperature excursion and period of an eddy are the drivers of heat flux. Calculating the instantaneous soil heat flux as a function of ground surface temperature and deriving sensible heat flux as a residual of the energy budget results in the third method for sensible heat flux calculation based on TIR imaging (Morrison et al. 2017), although in non-arid ecosystems it would be strongly dependent on the observations of both net radiation and latent heat flux.


**Conclusions**

The present study develops a framework for planetary boundary layer turbulence analysis based on UAS thermal camera measurements. The methods for thermal sequence retrieval, its post-processing and detection of large coherent structures were proposed. The performance and validity of the methods were tested in a case study over a flat and treeless boreal peatland in

South Finland. The spectral and morphological analysis pointed at the domination of large coherent structures up to tens of meters in width and hundreds in length, as expected in a convective PBL. Wind parameters independently observed by ground-based eddy-covariance setup provided support to the turbulence statistics derived by thermal sequence analysis. However, the novel 2D approach of this study also allowed for detection of instationary events such as counter-flow sweeps, which is beyond the capacity of previous observational methods. Larger, longer and more linear eddies were associated with lower instability

as expressed by the stability parameter $z\,L_O^{-1}$, while smaller and more circular eddies were observed at higher instability. The association between the surface temperature fluctuations on the ground and in the air, and the possibility to directly infer the residence time of an eddy and T fluctuation created by it, prepare ground for the application of sensible heat flux estimation by flux variance (Albertson et al. 1995) and surface renewal (Paw U et al. 1995) methods.

**Appendix A: Details of the vignetting correction**

The fourth-order polynomial describing the vignetting effect (see Fig. A1) is described by:

$S(x, y) = p_{0,0} + p_{1,0} \cdot x + p_{0,1} \cdot y + p_{2,0} \cdot x^2 + p_{1,1} \cdot x \cdot y + p_{0,2} \cdot y^2 + p_{3,0} \cdot x^3 + p_{2,1} \cdot x^2 \cdot y + p_{1,2} \cdot x \cdot y^2 + p_{0,3} \cdot y^3 + p_{4,0} \cdot x^4 + p_{3,1} \cdot x^3 \cdot y + p_{2,2} \cdot x^2 \cdot y^2 +$

$p_{1,3} \cdot x \cdot y^3 + p_{0,4} \cdot y^4$           (A1)

with x and y being the x and y- coordinate of an image pixel and $p_{i,j}$ the fit parameter value. The derived lens-specific coefficient

values are given in Table A1.

**Table A1.** Coefficients of the 4th degree polynomial surface describing the vignetting effect.

| Coefficient | value | 95% CI |
|---|---|---|
| $p_{0,0}$ | 23.79 | 23.78, 23.79 |
| $p_{1,0}$ | 0.02014 | 0.0201, 0.02019 |
| $p_{0,1}$ | 0.01298 | 0.01292, 0.01303 |
| $p_{2,0}$ | -5.922e-05 | -5.944e-05, -5.9e-05 |
| $p_{1,1}$ | -5.36e-05 | -5.382e-05, -5.338e-05 |
| $p_{0,2}$ | -3.069e-05 | -3.104e-05, -3.034e-05 |
| $p_{3,0}$ | 8.734e-08 | 8.687e-08, 8.782e-08 |
| $p_{2,1}$ | 8.291e-08 | 8.245e-08, 8.336e-08 |
| $p_{1,2}$ | 9.391e-08 | 9.334e-08, 9.449e-08 |
| $p_{0,3}$ | 2.667e-08 | 2.574e-08, 2.761e-08 |
| $p_{4,0}$ | -6.603e-11 | -6.639e-11, -6.567e-11 |
| $p_{3,1}$ | -3.4e-12 | -3.795e-12, -3.006e-12 |
| $p_{2,2}$ | -1.465e-10 | -1.47e-10, -1.46e-10 |
| $p_{1,3}$ | 4.615e-13 | -1.547e-13, 1.078e-12 |
| $p_{0,4}$ | -2.202e-11 | -2.29e-11, -2.114e-11 |

The vignetting correction matrix was obtained by subtracting the polynomial fit from the mean temperature of the center. We assumed that vignetting is zero at the center and computed the mean temperature from of the 40 x 40 pixel square zone at the center of the image. Finally, vignetting was eliminated by adding the correction matrix to each frame obtained in the field.

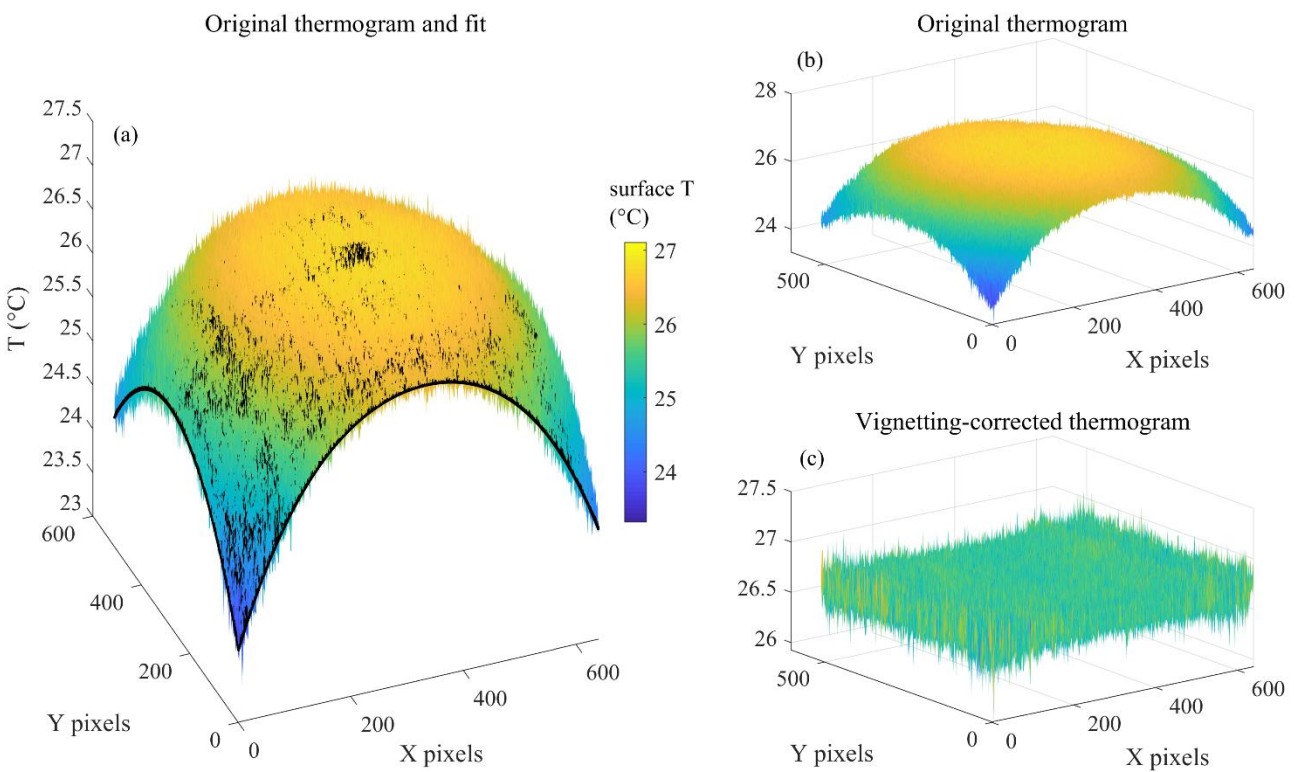

**Figure A1.** (a) 4th degree polynomial fit (black surface) and the thermogram of the black fabric (RMSE = 0.08 K); (b) uncorrected thermogram of black fabric surface (same data as in (a)); (c) vignetting-corrected thermogram.

**Appendix B: Camera Intrinsics**

The geometric calibration of the TIR camera was performed using a checkerboard so as to obtain the camera intrinsic parameters given in Table B1. Figure B1 gives a screenshot of the geometric calibration process in the Matlab® Camera Calibrator tool.

**Table B1.** Camera intrinsic parameters.

| Parameter | | |
|---|---|---|
| radial distortion | $k_1$=-0.049 | $k_2$=0.646 |
| tangential distortion | $p_1$=0.0035 | $p_2$=0.0028 |
| focal length | $f_x$=$f_y$=801 mm | |
| principal point | $c_x$=327 pix | $c_y$= 276 pix |
| skew | θ=3.4796 degrees | |
| mean reprojection error | 0.26 pixels | |

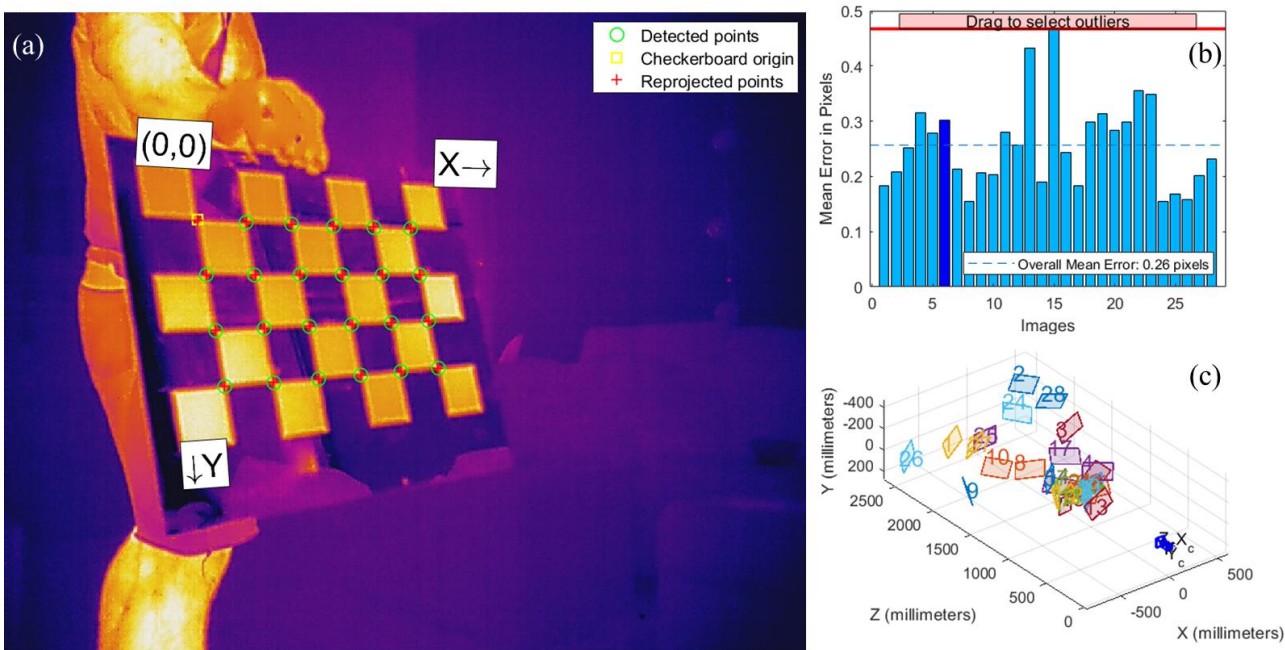

**Figure B1.** Screenshot from the Matlab® Camera Calibrator tool. (a) One of the 28 images used to derive the calibration parameters; (b) mean reprojection error (pix) for each image; (c) relative positions of the checkerboard relative to the camera.

**Appendix C: Image similarity metrics**

Structural Similarity Index (SSIM) is a measure of similarity between two images that simultaneously uses mean pixel values, pixel standard deviations and pixel cross-correlations, in order to assess the differences in luminance, contrast and image structure, respectively (Renieblas et al. 2017):

$$SSIM = \frac{1}{XY}\sum_{x=1}^{X}\sum_{y=1}^{Y}[l(x,y)]^{\alpha} \cdot [c(x,y)]^{\beta} \cdot [f(x,y)]^{\gamma} \tag{C1}$$

where X is the number of pixels in horizontal dimension, Y the number of pixels in the vertical dimension, x and y correspondingly the horizontal and vertical pixel coordinate, α, β and γ the positive constants. The terms $l_{ri}$, $c_{ri}$ and $f_{ri}$ stand for the luminance, contrast and image structure similarity between two images, respectively,

$$l_{ri} = (2\mu_r\mu_i + N1)/(\mu_r^2 + \mu_i^2 + N1) \tag{C2}$$

$$c_{ri} = (2\sigma_r\sigma_i + N2)/(\sigma_r^2 + \sigma_i^2 + N2) \tag{C3}$$

$$f_{ri} = (cov_{ri} + N3)/(\sigma_r\sigma_i + N3) \tag{C4}$$

where μ is the pixel mean of an image, σ the pixel value standard deviation, cov the covariance, N1, N2 and N3 are constants (Renieblas et al. 2017). The indices r and i stand for reference and sample image, respectively. SSIM was calculated using the Matlab® function *ssim*.

MSE and PSNR are standard tools for the assessment of similarity between a pair of images and were computed using Matlab® functions *immse* and *psnr*, respectively. Mean squared error is defined as (Gonzalez and Woods 1992):

$$MSE = \frac{1}{XY}\sum_{x=1}^{X}\sum_{y=1}^{Y}[g_r(x,y) - g_i(x,y)] \tag{C5}$$

where $g_r$ the reference image, $g_i$ the sample image.

PSNR is calculated as (Gonzalez and Woods 1992):

$$PSNR = -10log_{10}\frac{MSE}{S^2} \tag{C6}$$

where $S^2$ is the maximum pixel value.

**Appendix D: Georeferencing**

An inverse affine transformation was calculated for the reference frame of each flight to transfer it to the UTM35 co-ordinates, with the RMSE of fits equaling 0.32, 0.11, 0.48, 0.43 m for the flights 1–4, respectively:

$$UTM_{lat} = -(a·f - a·y_{pix} - c·d + d·x_{pix}) / (a·e - b·d) \tag{D1}$$

$$UTM_{lon} = (-e·c + e·x_{pix} + f·b - b·y_{pix}) / (a·e - b·d) \tag{D2}$$

where a, b, c, d, e, f are the parameters of affine transform, $x_{pix}$ and $y_{pix}$ the (right-handed) coordinates of a pixel in an original image, and $UTM_{lat}$ and $UTM_{lon}$ the UTM northing and easting of a transformed image. As the resolution of the original images was about 0.6 m, for convenience, a 1 m resolution was used for the UTM grid onto which the images were transferred. As the images were co-registered in Step 3 (Sect. 2.2.3), the same set of parameters was applied to georeference each subsequent image of a sequence.

**Table D1.** Affine transform coefficients applied to the registered thermogram sequences of each flight along with the root-mean-squared values (RMSE).

|          | flight 1<br>6 August | flight 2<br>6 August | flight 3<br>28 August | flight 4<br>28 August |
|----------|------------|------------|------------|------------|
| a        | -0.2536    | -0.2490    | -0.1683    | -0.2573    |
| b        | -0.6017    | -0.6057    | -0.6430    | -0.6124    |
| c        | 352435.98  | 352431.41  | 352404.79  | 352425.36  |
| d        | 0.6009     | 0.6048     | 0.6430     | 0.6124     |
| e        | -0.2533    | -0.2492    | -0.1659    | -0.2576    |
| f        | 6858603.63 | 6858595.52 | 6858569.46 | 6858599.99 |
| RMSE (m) | 0.32       | 0.11       | 0.48       | 0.43       |


## Code availability

The codes are freely available at https://doi.org/10.5281/zenodo.4019155.


## Data availability

The data are freely available at https://doi.org/10.5281/zenodo.4019321.

## Video supplement

The visualizations of turbulence for the four flights are available at https://doi.org/10.5281/zenodo.4019175.


## Video supplement link

Additionally, the videos may be watched on YouTube, Flight 1, 6 August: https://youtu.be/UwN8rFQ3Y0E, Flight 2, 6 August: https://youtu.be/UeNU8lq7krY, Flight 1, 28 August: https://youtu.be/K4ahj0EtrWM, Flight 2, 28 August: https://youtu.be/jgC2GDptLtU.


## Author contributions

Dr. Pavel Alekseychik conceived the study, conducted the measurements and their analyses, wrote the text and produced the figures. Prof. Gabriel Katul helped develop the theoretical and analytical framework and contributed to the text. Dr. Ilkka Korpela helped develop the image processing methods and contributed to the text. Dr. Samuli Launiainen made major

contributions throughout the study including theoretical and analytical methods and writing parts of the text.

## Competing interests

The authors declare no competing interests.


## Acknowledgements

PA and SL acknowledge the support of the projects CLIMOSS (*Climate impacts of boreal bryophytes - from functional traits to global models*) funded by the Academy of Finland, Decision no. 296116), and SOMPA *(Novel soil management practices - key for sustainable bioeconomy and climate change mitigation*, funded by the Strategic Research Council at the Academy of Finland, Decision no. 312912). Prof. Timo Vesala (INAR, University of Helsinki) is gratefully acknowledged for providing

the drone and thermal camera for the use in this study.

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
