# Peer review of "Eddies in motion: visualizing boundary-layer turbulence above an open boreal peatland using UAS thermal videos"

_Atmospheric Measurement Techniques, 2020_

## Referee Comment (RC1) · Anonymous Referee #1 · 16 Nov 2020

General Comments:

The article uses a thermal imager on a UAS flown at 500 m over boreal peatland to extract surface temperature fluctuation using methodology from Christen et al 2012. Several corrections are applied to the dataset to allow accurate assessment of turbulent statistics. From the temperature fluctuations TIV is performed to extract 2D velocity fields and a spectral analysis is performed to identity eddy size and shape for eddies of the order of 10-100 m. Further analysis looking at the size and aspect ratio of the eddies is compared to relevant turbulent statistics. I applaud the authors on their methodology and problem-solving skills to treat this dataset to bring it to the

point where turbulence analysis can be performed. I too have dealt with many of the issues described here without any publication to speak of. I see the science within this document as a reflection as step forward for UAS-TIR measurements in atmospheric boundary layer research. While the results and methodology of the document are worthwhile of publication, I have provided a series of major and minor comments below. I find the merit and methodology of the work very high, but the presentation and text to only be fair, thus the comments accumulate to a minor revision. Primarily, I find that the manuscript is lacking a strong review of the literature. I recommend the authors spend more time in the introduction and text to provide citations on turbulent structures. Specifically, the LES community has been studying this topic for some time and should be included. Additionally, the text is very informal, often using inconsistent abbreviations and acronyms. The authors should re-read and work the text to improve its quality and consistency.

Major Comment: Line 53: Be careful, throughout the document you interchangeably use IR and TIR. I understand the overlap, but IR is reserved for short-wave while TIR is reserved for mid-to-long wave infrared radiation. I am only going to comment here on this but will need to change throughout the document.

Major Comment: How did you handle the transmissivity? At such an altitude (500 m) there is likely some degree of error introduced from the transmissivity of the air on the accuracy of the measurement. I believe the FLIR software has a default correction based on the air temperature and humidity, did you use this to correct for the transmissivity?

Major Comment: Many informal sentences. Please use input from co-authors or a reputable grammar editor like Grammarly to help improve make the text more formal. Please rewrite any sentence with the word "so" or beginning with the word "Because." Such phrasing leads to informal sentences.

Minor comment: What is the anticipated error from the noise introduced leaves? Are

there leaves? How does the surface look? A high quality (larger than Figure 4) visual image or satellite view would be very helpful.

Minor comment: Can you add a photo of the flux tower setup?

Minor comment: Please provide more details on how the imregister function (as well as other functions) work. Please remember that Matlab is a paid programing language such that the methodology should be explained as to someone is reprograming this methodology with another language like C++.

Minor Comments:

Line 15: comma after (UAS)

Line 19: Please change "whilst" to "while"

Line 23: The UAS thermal imagery is collocated with a ground-based eddy-covariance system.

Line 45: Remove "made"

Line 50: Replace regretted with reported

Line 58: Remove "Evidently"

Line 67: Replace 2-day to two-day

Line 70: Rewrite for clarity: "..and the available eddy-covariance (EC) tower..."

Line 75: Please include the spectral response of your camera

Line 82: Remove "quite well"

Line 82-83: Rewrite

Line 91: Replace ".,." with a period "."

Line 91: Can you comment on the time synchronization more? For long averaging

periods (>5 minutes) it may not be a concern, but for detection of large eddies this is rather important. Was this method with the aluminum sheet synchronized with a watch? Was there on an onboard GPS available? Was the EC tower GPS synchronized.

Line 93: Remove "easily"

Line 98: Please provide a literature source for the emissivity value used.

Line 108 and 109: Remove "The" in phrase "The Steps," also "Steps" is not capitalized

Figure 2: Please add a more informative caption.

Line 129: Rewrite 129 to not begin with "Because"

Line 141: Remove "To do that"

Line 161: What is a deviation from a space-time average? This doesn't make sense to me.

Line 162: Replace "so" with "such"

Line 165: No need for "e.g."

Line 169: This is confusing to me. A forward finite difference already implies it was divided the time. Was dT/dt multiplied by the sampling frequency after this?

Line 173: Remove "now"

Line 181: Rewrite to "A 2D wavelet..."

Line 181: Remove "then"

Line 186: Incomplete sentence "The positive..."

Line 194: Please spell out wind direction or define WD

Lines 205-210: Here you are calling the methodology PIV. While it is true you are borrowing methodology from PIV, the community has adopted the terminology Thermal

Image Velocimetry (TIV) when using "thermal" particles.

Line 215: Replace "that is" with "as"

Line 216: (WS)

Line 218: u star, z_0 and L should all be in parentheses.

Line 220: I think partitioned should be replaced with temporally averaged.

Line 224: Informal sentence, please rewrite

Line 225: Here you abbreviate August. Please spell it out like you did earlier in the document.

Line 227: "3-m"

Table 2: Is z L0 suppose to be z/L0?

Caption Figure 4: Here you use a percentage for the emissivity. Earlier in the document you use a factional number. Please be consistent.

Line 258: Remove "probably" and please hedge this sentence more formally.

Line 271: Super interesting about the heat capacity of the needles!

Line 279: Spell out north and south

Line 281: Please define sigma. I assume you are talking about the standard deviation.

Line 286: When was this spectrum taken? Using which data? Can you comment here on the difference between the signals at the larger frequencies. I think this is a interesting result.

Line 305: Please spell out temperature

Line 342: Please change "power" to "spectral power density"

Line 349: Remove "However"

Line 350: Change the colon to a comma

Line 358: Please change "additionally explored by inquiry" to "studied by dividing the signal"

Line 369: Remove ",too,"

Figure 10: Please add a legend for the red lines

Line 371: Please correct for informalities.

Figure 10: It is interesting to see the larger variability in the ratio of the 128-cross and along wind structures peak for flight 3. I understand this to be the flight with the fanning pattern observed.

Line 395: Can you rewrite, I don't understand what you mean "flights 1-2 group close together…" and so forth

Figure 11: What's the major and minor axis?

Line 423: What do you mean by "associated"?

Line 424: Please use consistent nomenclature for "R2"

Figure 13: It would be easier to interpret this one to one if the limits of the axes were the same.

Figure 13: Which footprint methodology did you use? Please cite.

Line 434: Change from "Such,.." to "The"

Line 436: Please use a comma instead of the colon

Line 440: These are indeed "large" structures but are not the "largest structures." I would be more specific here and say structures ranging from 1-420 m structures.

Line 443: Again, be careful here about how you talk about turbulent length scales. The smallest scales of turbulence are order 1 mm.

Lines 440-452: While I agree this method is very advantageous and progressive, some of the previous works mentioned were looking at smaller scale turbulence. For the goal of looking at TOS I agree a larger field of view from a UAV is perfect, but the tradeoffs were looking resolution for smaller scale processes and sensitivity from using a thermal imager with a microbolometer.

Line 464: Plethora is informal

Line 470: Add citations here about wind speed and TOS. Such papers as "Surface Thermal Heterogeneities and the Atmospheric Boundary Layer: The Relevance of Dispersive Fluxes" by Margairaz et al and "Buoyancy effects on the integral length scales and mean velocity profile in atmospheric surface layer flows" by Salesky et al.

Lines 434-500: I do not feel like this discussion is anything more than a conclusion of the presented work.

Lines 490-495: Several methods exist to exact SHF from thermal imaging products. Morrison et al 2012 as well as other remote sensing papers should be discussed here.

---

## Referee Comment (RC2) · Anonymous Referee #2 · 10 Feb 2021

Dear Authors, This study used the aerial thermal imaging to detect the turbulence characteristics near the ground surface. It is compared with the ground-based sonic anemometer measurement in an EC site. Although the time-sequential TIR imaging has already been used for the same purpose, the use of UAS to capture wider area has not been accomplished before and its knowledge is useful. Meanwhile, there are some points to be clarified especially in the data analysis to keep the generality of the discussion.

Comments: L91 Delete ",."

L134 Please explain what is the Structural Similarity Index. Also, how are the RMSE

and SNR defined in this process?

L181 Please explain why the authors selected 150 m in this process.

L182, L185 Please show how sensitive these parameters (14 m and +-3.5 m) for the latter discussion (i.e. the ratio of the length and width of the isolated structures, Figs.11 and 12).

P210 Please describe how large the interrogation area in meters, and also the time increment to derive the velocity in sec.

P212 Please describe the mean height of the roughness elements (vegetations) of the observation area.

P219 How is the flux footprint used in the latter analysis and/or discussion?

L285 Is this FFT analysis applied for the time series of the surface temperature at a certain point in the images, and later it is averaged horizontally? Is there any reason why the two spectra in Fig,7 are different at the low frequency region? Are there -1 power law region (e.g. Drobinski et al. 2004) both in the spectra of EC Ts and UAV Tg?

L308 "The relatively small ...." It is difficult to understand this sentence just from the corresponding figures (Fig.8a,b).

L318 "Wall effects at the forest edge ..." This is not certain yet from the snapshots of the temperature anomaly. It should be evaluated, for example, after ensemble or temporal average to extract the effect of the heterogeneous roughness.

Figure 8 Is there any extra process to obtain these velocity vectors after the image correlation calculation? Please describe details about it if there are any (i.e. smoothing, averaging, handling of the error vectors, etc.). Also, please describe how the result of PIV calculation is sensitive to the accuracy of the image registration and/or georeferencing.
L335 "the EC WS was higher ..." This is interesting since the movement of the surface temperature structures seems to be associated with the convective thermal structures in this observation, which probably move faster than the bottom air whose speed is measured by EC (z=3m, below RSL) if the mean wind profile follow the typical log-law plus MOS function. Please explain why EC WS is faster. Some discussion were seen in Garai et al (2013) and Inagaki et al. (2013).

Figure 9 Are the periods of the lower wavelet power, which are the majority of the entire period, corresponding to the quiescent period as in Fig.8b? Please describe what happens in it.

L365 Probably, the spectral powers at 128m and 10m are selected due to the FOV and the resolution of the observation. Are they representing the entire spectral shapes? Please describe, for example, they are within the energy containing range or the inertial subrange if those wavenumber spectra follow the ordinal spectral shape of turbulence.

L427 "... were contemporaneous with ..." Does this mean that 5-min average is not enough long relative to the time scale of the large coherent structures?

An extra comment. This study is motivated to examine the applicability of the TIR imaging for the surface heat flux measurement as written in the entire of the manuscript. It is also obviously written in the last section. Besides, there is no direct comparison between the ground-based sensible heat flux and the TIR images. Therefore, I recommend to add the data of the sensible heat flux together with that of TIR (e.g. show together with Fig.9,12,13).

References: Drobinski P, Carlotti P, Newsom RK, Banta RM, Foster RC, RedelspergerJ-L (2014) The Structure of the Near-Neutral Atmospheric Surface Layer. J Atmos Sci 61(6), 699–714. Garai A, Pardyjak E, Steeneveld G-J, Kleissl J (2013) Surface Temperature and Surface-Layer Turbulence in a Convective Boundary Layer. Boundary-Layer Meteorol, 148, 51–72. Inagaki A, Kanda M, Onomura S, Kumemura H (2013) Thermal Image Velocimetry. Boundary-Layer Meteorol, 149, 1–18.

---

## Author Comment (AC1) · 10 Mar 2021

**Response to Review 1**

Anonymous Referee #1

General Comments: The article uses a thermal imager on a UAS flown at 500 m over boreal peatland to extract surface temperature fluctuation using methodology from Christen et al 2012.

Several corrections are applied to the dataset to allow accurate assessment of turbulent statistics. From the temperature fluctuations TIV is performed to extract 2D velocity fields and a spectral analysis is performed to identity eddy size and shape for eddies of the order of 10-100 m. Further analysis looking at the size and aspect ratio of the eddies is compared to relevant turbulent statistics. I applaud the authors on their methodology and problem-solving skills to treat this dataset to bring it to the point where turbulence analysis can be performed. I too have dealt with many of the issues described here without any publication to speak of. I see the science within this document as a reflection as step forward for UAS-TIR measurements in atmospheric boundary layer research. While the results and methodology of the document are worthwhile of publication, I have provided a series of major and minor comments below. I find the merit and methodology of the work very high, but the presentation and text to only be fair, thus the comments accumulate to a minor revision. Primarily, I find that the manuscript is lacking a strong review of the literature. I recommend the authors spend more time in the introduction and text to provide citations on turbulent structures. Specifically, the LES community has been studying this topic for some time and should be included. Additionally, the text is very informal, often using inconsistent abbreviations and acronyms. The authors should re-read and work the text to improve its quality and consistency.

- Thank You very much for the very positive evaluation of this work and providing many useful comments. We have made the requested improvements and edits throughout the text.

Major Comment: Line 53: Be careful, throughout the document you interchangeably use IR and TIR. I understand the overlap, but IR is reserved for short-wave while TIRis reserved for mid-to-long wave infrared radiation. I am only going to comment hereon this but will need to change throughout the document.

- All instances of IR were replaced with TIR.

Major Comment: How did you handle the transmissivity? At such an altitude (500 m)there is likely some degree of error introduced from the transmissivity of the air on the accuracy of the measurement. I believe the FLIR software has a default correction based on the air temperature and humidity, did you use this to correct for the transmissivity?

- That is correct, we entered the 500 m distance and the measured RH and Tair in the ResearchIR software when processing the thermal data, which provided a robust estimate of the atmospheric effect. A small error in the transmissivity assessment, even if present, would not significantly affect the results of this study – it would mainly affect the absolute level of the temperatures, while temperature fluctuations are mainly discussed. A future study may attempt to calculate a time-varying change in transmissivity due to the passage large eddies with different properties.

Major Comment: Many informal sentences. Please use input from co-authors or a reputable grammar editor like Grammarly to help improve make the text more formal. Please rewrite any sentence with the word "so" or beginning with the word "Because." Such phrasing leads to informal sentences.

- Thank You for pointing this out, we tried to improve the language throughout the manuscript.

Minor comment: What is the anticipated error from the noise introduced leaves? Are there leaves? How does the surface look?

- the surface is effectively flat, as the microtopography is quite undeveloped (elevation difference between hummocks and hollows <15 cm). The vegetation is short-stature, being mainly represented by sedges and shrubs. The combined leaf area index of the sedges and shrubs was about 0.4 on the measurement days, which is a low figure typical of open fens. While the sedges contribute the most to the site-average LAI, their stems are thin and vertically oriented (before the start of senescence), so that when observing the peatland in a top-down perspective of the UAV, one sees mainly the moss surface. The ground photo (new Figure 1c) may be misleading in that sense, as at this angle, the sedges have a much greater projected area. So the temperatures presented in the manuscript are largely those of the moss cover.
- Also, no strong wind was recorded on the measurement days. Therefore, the expected magnitude of thermal noise resulting from wind-induced leaf flapping the is only minor, and it gets further reduced by the spatial averaging over a 1x1 m grid.

A high quality (larger than Figure 4) visual image or satellite view would be very helpful.
Minor comment: Can you add a photo of the flux tower setup?

- Both maps have been added (new Figure 1).

Minor comment: Please provide more details on how the imregister function (as well as other functions) work. Please remember that Matlab is a paid programing languagesuch that the methodology should be explained as to someone is reprograming this methodology with another language like C++.

- Done. However, I am positive that other programming environments offer analogous functionality using the same (or similar) approaches.

Minor Comments:
Line 15: comma after (UAS) - Done
Line 19: Please change "whilst" to "while" - Done
Line 23: The UAS thermal imagery is collocated with a ground-based eddy-covariancesystem. - Done
Line 45: Remove "made" - Done
Line 50: Replace regretted with reported - Done
Line 58: Remove "Evidently" - Done
Line 67: Replace 2-day to two-day - Done
Line 70: Rewrite for clarity: "..and the available eddy-covariance (EC) tower..." - Done
Line 75: Please include the spectral response of your camera - Done

Line 82: Remove "quite well" - Done

Line 82-83: Rewrite - Done

Line 91: Replace ".," with a period "." - Done

Line 91: Can you comment on the time synchronization more? For long averaging periods (>5 minutes) it may not be a concern, but for detection of large eddies this is rather important. Was this method with the aluminum sheet synchronized with a watch? Was there on an onboard GPS available? Was the EC tower GPS synchronized. – the corresponding explanations were added. The drone had an onboard GPS; the EC coordinates are known from a measurement with a survey-grade GPS.

Line 93: Remove "easily" - Done

Line 98: Please provide a literature source for the emissivity value used. – Unfortunately, as literature values do not seem to be available for *Sphagnum* moss (or effective moss-dominate ecosystem) emissivity, we are forced to work with the generic value of 0.98. This value of 0.98 is chosen based on expectation that highly moisturized capitulum of Sphagnum has a high emissivity approaching 99%. Leaf emissivities of deciduous species are about 98%, e.g. Kim et al. 2012.

Line 108 and 109: Remove "The" in phrase "The Steps," also "Steps" is not capitalized  - Done

Figure 2: Please add a more informative caption. - Done

Line 129: Rewrite 129 to not begin with "Because"  - Done

Line 141: Remove "To do that" - Done

Line 161: What is a deviation from a space-time average? This doesn't make sense to me. – this is the overall mean temperature of the entire flight, obtained by first averaging the temperatures within each image, and then averaging those over the whole recorded sequence. Each images is essentially adjusted so that to make its average equal to that "space-time average". As explained in the text, this is done to alleviate the calibration drift of the sensor.

Line 162: Replace "so" with "such" - Done

Line 165: No need for "e.g." - Done

Line 169: This is confusing to me. A forward finite difference already implies it was divided the time. Was dT/dt multiplied by the sampling frequency after this? – I apologize for the inconsistency – this quantity is actually not used in the current version of the manuscript. This sentence is now removed.

Line 173: Remove "now"  - Done

Line 181: Rewrite to "A 2D wavelet..." - Done

Line 181: Remove "then" - Done

Line 186: Incomplete sentence "The positive..." – What do you mean? I think the sentence is complete, "The positive and negative regions remaining after that filtering operation represent, in essence, the smoothed boundaries of the larger coherent structure thermal traces"

Line 194: Please spell out wind direction or define WD - Done

Lines 205-210: Here you are calling the methodology PIV. While it is true you are borrowing methodology from PIV, the community has adopted the terminology Thermal Image Velocimetry (TIV) when using "thermal" particles. – we agree with that, PIV changed to TIV throughout the document.

Line 215: Replace "that is" with "as" - Done

Line 216: (WS) - Done

Line 218: u star, z_0 and L should all be in parentheses. - Done
Line 220: I think partitioned should be replaced with temporally averaged. - Done
Line 224: Informal sentence, please rewrite - Done
Line 225: Here you abbreviate August. Please spell it out like you did earlier in the document. - Done
Line 227: "3-m" - Done
Table 2: Is z L0 suppose to be z/L0? – that is correct, this is z/Lo written in exponential notation. Perhaps it wasn't correctly presented in the document for you.
Caption Figure 4: Here you use a percentage for the emissivity. Earlier in the document you use a factional number. Please be consistent. - Done
Line 258: Remove "probably" and please hedge this sentence more formally. - Done
Line 271: Super interesting about the heat capacity of the needles! – thank You for mentioning this - indeed a factor to be accounted for. The forest canopy temperature does fluctuate much more than the moss (or any other sparsely vegetated surface) due to the vast difference in heat capacity and atmospheric coupling (the latter has been added to the text, as it is an equally important aspect controlling the heat exchange with the air.)
Line 279: Spell out north and south - Done
Line 281: Please define sigma. I assume you are talking about the standard deviation. - Done
Line 286: When was this spectrum taken? Using which data? Can you commenthere on the difference between the signals at the larger frequencies. I think this is ainteresting result.- added "The generally lower spectral energy of the UAS $T_s$ data is due to the fact that the high heat capacity of the ground leads to much lower surface temperature fluctuations than those observed in the airflow. The flattening of the UAS spectrum at higher frequencies results from noise contributed mainly by the thermal measurement and the image registration error."
Line 305: Please spell out temperature - Done
Line 342: Please change "power" to "spectral power density" - Done
Line 349: Remove "However" - Done
Line 350: Change the colon to a comma - Done
Line 358: Please change "additionally explored by inquiry" to "studied by dividing the signal" - Done
Line 369: Remove ",too," - Done
Figure 10: Please add a legend for the red lines - Done
Line 371: Please correct for informalities. – Done
Figure 10: It is interesting to see the larger variability in the ratio of the 128-cross and along wind structures peak for flight 3. I understand this to be the flight with the fanning pattern observed. – Actually, this was the flight with elongated linear structures. I can interpret the high peaks in the ratio of the 128m along:cross spectral powers as the periods when such structures were at their peak development. It is a little counter-intuitive, but it seems that the highest power occurs when the entire field of view is occupied by a burst of structures approaching the extent of FOV in the given direction (ca. 300-400m long). Thus, a packet of long, intense linear structure creates a greater spectral 128m power in the along-wind direction than a packet of shorter structures would.
Line 395: Can you rewrite, I don't understand what you mean "flights 1-2 group closetogether..." and so forth - Done

Figure 11: What's the major and minor axis? – Added the explanation: Major axis is the greater dimension of the coherent structure's thermal trace, which is always oriented in streamwise direction; correspondingly, the minor axis is the "width".
Line 423: What do you mean by "associated"? - Done
Line 424: Please use consistent nomenclature for "R2" - Done
Figure 13: It would be easier to interpret this one to one if the limits of the axes werethe same. – This is maybe unnecessary as different quantities are presented in (a-c).
Figure 13: Which footprint methodology did you use? Please cite. – it was Kormann and Meixner (2001). Citation added.
Line 434: Change from "Such,.." to "The" - Done
Line 436: Please use a comma instead of the colon - Done
Line 440: These are indeed "large" structures but are not the "largest structures." I would be more specific here and say structures ranging from 1-420 m. - Done
Line 443: Again, be careful here about how you talk about turbulent length scales. The smallest scales of turbulence are order 1 mm. - Done
Lines 440-452: While I agree this method is very advantageous and progressive, some of the previous works mentioned were looking at smaller scale turbulence. For the goal of looking at TOS I agree a larger field of view from a UAV is perfect, but the tradeoffs were looking resolution for smaller scale processes and sensitivity from using a thermal imager with a microbolometer. – We quite agree with this. Hopefully, future research will succeed in minimizing the artefacts of the thermal camera data and the processing methods, and improve the quality of the derived turbulence characteristics!

Line 464: Plethora is informal - Done

Line 470: Add citations here about wind speed and TOS. Such papers as "Surface Thermal Heterogeneities and the Atmospheric Boundary Layer: The Relevance of Dis-persive Fluxes" by Margairaz et al and "Buoyancy effects on the integral length scales and mean velocity profile in atmospheric surface layer flows" by Salesky et al. – Thank You so much for the suggestions, they have been added. Our findings find support in this literature, indeed.

Lines 434-500: I do not feel like this discussion is anything more than a conclusion of the presented work. – Perhaps, but these are the points we felt can be raised based on the proof-of-concept study that had been conducted. A larger dataset (some tens of flights spanning the range of stabilities) would allow for a deeper analysis. We hope to undertake such an effort in the future.

Lines 490-495: Several methods exist to exact SHF from thermal imaging products.Morrison et al 2012 as well as other remote sensing papers should be discussed here. – Thank You for the suggestion, this is added. However, in this approach the sensible heat flux is a residual of the energy budget, and typically the smallest component of it in boreal peatlands. In such ecosystems, the surface distribution of LE would be an important factor controlling the local-scale (ca. 1 m) energy budget, and thus introduce a strong uncertainty. This can be avoided by measuring over fallow agricultural fields during drought. However, if the turbulence over a given ecosystem at given conditions must be addressed, these uncertainties inevitably have to be dealt with.

---

## Author Comment (AC2)

**Response to Review 2**

Dear Authors, This study used the aerial thermal imaging to detect the turbulence characteristics near the ground surface. It is compared with the ground-based sonic anemometer measurement in an EC site. Although the time-sequential TIR imaging has already been used for the same purpose, the use of UAS to capture wider area has not been accomplished before and its knowledge is useful.

- Thank You, we deeply appreciate this positive response. While the current study is rather experimental, we hope that it lays down the foundations for a new generation of turbulence studies. Meanwhile, there are some points to be clarified especially in the data analysis to keep the generality of the discussion. – We did our best to address the points You have raised.

**Comments: L91 Delete ",." - Done**

L134 Please explain what is the Structural Similarity Index. Also, how are the RMSE and SNR defined in this process? – Thank You for the question – detailed explanations have been added as new Appendix C.

L181 Please explain why the authors selected 150 m in this process. – The document must have been incorrectly presented on your screen; the text actually reads "at the spatial scales

of 1-50 m". So the wavelet analysis was done for those spatial scales. However, the explanation is now added, such a range was chosen based on the assumption that the maximum eddy size well represented by a TIR image cover"ing ca. 300x400 m would be roughly 100 m."

L182, L185 Please show how sensitive these parameters (14 m and  $\pm 3.5$  m) for the latter discussion (i.e. the ratio of the length and width of the isolated structures, Figs.11and 12). – Thank You for raising this important point. Indeed, they introduce some sensitivity in the processing and need to be treated with care. The wavelet decomposition scale is more sensitive than the filtering threshold. The following comments were added: "It must be noted that the wavelet transform scale is a sensitive parameter requiring adjustment to the scale of the dominant eddies; an excessively small scale value would lead to improper fractionation of the eddies, while a scale value which is too high would result in grouping where the eddies are apparently separate.", and "The threshold for this filtering operation should also be chosen with care, as the slopes separating positive and negative wavelet regions can be steep (see the effect of the  $\pm 3.5$  threshold in Fig. 3c)."

L210 Please describe how large the interrogation area in meters, and also the timeincrement to derive the velocity in sec. – The resolution of the georeferenced images is 1m/pix, therefore the interrogation area was 100 m; as per the specifics of the PIV method, the sequence is divided into "frames", i.e. pairs of images, and the wind field is computed for each frame before they are averaged for the given periods of interest – as a result, one can say that the time increment is 2s.

P212 Please describe the mean height of the roughness elements (vegetations) of the observation area. – Description added. It is mainly the sedges that create nearly all of the roughness at the site; they grow to the mean height of about 0.25m.

P219 How is the flux footprint used in the latter analysis and/or discussion? – explanation has been added. EC footprint was used solely for reconciling the UAG surface temperatures with the sonic temperature.

L285 Is this FFT analysis applied for the time series of the surface temperature at a certain point in the images, and later it is averaged horizontally? – This is exactly correct, that's the ways the UAS FFT spectrum has been obtained. An explanation has been added in the end of section 2.2.6: "FFT was also applied to the thermal sequences in the temporal domain thus: first, FFT was performed on individual pixel time series, and those pixel-wise spectra were averaging to yield a single FFT spectrum of a flight."

Is there any reasonwhy the two spectra in Fig,7 are different at the low frequency region? – I think this is the region where the spectra are not representative of the turbulent fluctuation as they are calculated from short ( $\leq 20$  min) records. The difference increases at frequencies lower than 1e-4 Hz, to judge from the figure, which corresponds to a period comparable to the length of the record.

Another possibility lies in the contribution of poorly understood artifacts in the thermal data which the present methods failed to eliminate; those should be addressed in the future studies. However, I don't have any good answer as to what these may be.

Are there -1power law region (e.g. Drobinski et al. 2004) both in the spectra of EC Ts and UAVTg? – we did specifically attempt to detect the -1 power law, given its importance for the interpretation of turbulence origin, but the evidence remained inconclusive. Maybe the large-scale scturcture of the ABL during the flights did not favor the turbulent organization which leads to the -1 power law relationship.

L308 "The relatively small..." It is difficult to understand this sentence just from the corresponding figures (Fig.8a,b). -I would say 5-50 m based on visual inspection. In any case, the main point here is the contrast between the different regimes, which is rather apparent from the images of Fig.8.

L318 "Wall effects at the forest edge..." This is not certain yet from the snapshots of the temperature anomaly. It should be evaluated, for example, after ensemble or temporal average to extract the effect of the heterogeneous roughness. – Actually, the forest edge is most pronounced in the temperature standard deviation (Figure 6). I believe that this approach for visualizing the stepchange in surface roughness is analogous to what You are proposing.

Figure 8 Is there any extra process to obtain these velocity vectors after the image correlation calculation? Please describe details about it if there are any (i.e. smoothing, averaging, handling of the error vectors, etc.). – Done (added in 2.2.7). Also, please describe how the result of PIV calculation is sensitive to the accuracy of the image registration and/or georefer-encing. – This is an important question, now answered in 2.2.7.

L335 "the EC WS was higher..." This is interesting since the movement of the surfacetemperature structures seems to be associated with the convective thermal structures in this observation, which probably move faster than the bottom air whose speed is measured by EC (z=3m, below RSL) if the mean wind profile follow the typical log-law plus MOS function. Please explain why EC WS is faster. Some discussion were seenin Garai et al (2013) and Inagaki et al. (2013).

- This is a good point, thank You for raising it. Due to the uncertainty in the PIV process, the PIV "flow" velocity has a random (and possibly systematic?) uncertainty which can be estimated at 30%. It is also important to bear in mind that the specific input for the PIV was the wavelet transform at the scale of 5m, hence the PIV output shows the velocity field of the smaller eddies 5-10 m in size. Some previous research (this is now summarized in the updated Introduction) indeed found coherent structures to advect faster than the mean wind near the ground, but in the case of smaller eddies which are attached to the surface and well-coupled with the ground roughness, can well be advected at roughly the mean wind speed measured by the EC.

Figure 9 Are the periods of the lower wavelet power, which are the majority of theentire period, corresponding to the quiescent period as in Fig.8b? Please describe what happens in it. – Precisely so, the low wavelet powers (the bluish colours) in Fig. 9 correspond to such "quiescent periods". Large, well-defined structures contribute the most to the wavelet powers (especially at scales approaching the limit of 50 m), so when they were absent, the wavelet power dropped. A clarifying sentence has been added to the discussion of Fig.9.

L365 Probably, the spectral power s at 128m and 10m are selected due to the FOV and the resolution of the observation. Are they representing the entire spectral shapes? Please describe, for example, they are within the energy containing range or the inertial subrange if those wavenumber spectra follow the ordinal spectral shape of turbulence. – You are absolutely correct. 10 m is the limit is the eddy size which is well-represented in the measurements, and relevant for the coherent structure discussion; 128 m is the largest scale that still fits in the FOV. Undoubtedly, there are larger coherent structures (i.e. VLSMs) some kilometers long which fall outside the domain achievable in the current experiment. However, VLSMs are not anymore "eddies" in the classical sense of the word, and I think it would be fair to say that the scales represented in the current UAS experiment do illustrate the eddies on all relevant scales.

Regarding the eddy scales: the 128 m-scale structures have a characteristic length scale of  $\sim 10^{-2}$  Hz, i.e. correspond to the energy containing subrange (Fig. 7). Consequently, the smaller eddies (somewhere under 100 m in size) fall in the inertial subrange. This has been added to the discussion.

L427 "...were contemporaneous with..." Does this mean that 5-min average is notenough long relative to the time scale of the large coherent structures?

- This is a difficult question, but I'm grateful that You have raised it as it leads to some interesting discussion. In general, I have to agree that the 5 min averaging may not be the best approach as it leads to the loss of low-frequency contributions. It is mainly the shortness of the data set used in this study that prompted the division into 5 min periods, and had the data been more extensive, we would have used 30 min averages., Many similar studies use 5 min averages and probably suffer from similar issues. The simplest "back of the envelope" estimate whether the 5 min averaging is valid is by looking at the scalograms in Fig.9, dividing the period into four or two parts (for the short fourth flight). It appears that 5-min periods include several cycles of "intense" and "quiet" turbulence in the flights 3 and 4, a little less in the flight 1, and even less not in the flight 2. Perhaps longer averaging times would reduce the scatter in the relationships such as in Fig. 13 – this remains to be seen in a future study.

An extra comment. This study is motivated to examine the applicability of the TIR imag-ing for the surface heat flux measurement as written in the entire of the manuscript. It also obviously written in the last section. Besides, there is no direct comparison between the ground-based sensible heat flux and the TIR images. Therefore, I recommend to add the data of the sensible heat flux together with that of TIR (e.g. showtogether with Fig.9,12,13).

- Thank You for the suggestion. I have added the panels with the kinematic sensible heat flux in Figure 13, which seemed most suitable for these data. It seems that it would be wrong to expectat a simple link between the EC fluxes and the surface temperature, as the relationship is rather scattered, although there is a definite positive slope. 30-min averaging instead of 5-min may help eliminate some of this scatter (as discussed above), while a larger number of flights will further increase the R2.

References: Drobinski P, Carlotti P, Newsom RK, Banta RM, Foster RC,RedelspergerJ-L (2014) The Structure of the Near-Neutral Atmospheric Surface Layer.J Atmos Sci 61(6), 699–714.

Garai A, Pardyjak E, Steeneveld G-J, Kleissl J (2013)Surface Temperature and Surface-Layer Turbulence in a Convective Boundary Layer.Boundary-Layer Meteorol, 148, 51–72.

Inagaki A, Kanda M, Onomura S, KumemuraH (2013) Thermal Image Velocimetry. Boundary-Layer Meteorol, 149, 1–18.

---

## Referee Report (RR1)

Dear Authors,

Thank you for taking my and the other reviewer's comments and suggestions with care and addressing them one by one. I only have a few minor technical edits/suggestions that are provided below. Provided these technical edits are address I accept this paper for publication. Thank you.

Technical Edits:

Line 32: "sizes" to "size".

Line 33: "durations" to "duration".

Line: 34 remove "now".

Line 37: remove "already"

Line 46: "depend" to "depends".

Line 100: "of" to "is a"

Line 123: What do you mean by "decimated"? Did you cherry pick every 30$^{th}$ image, or did you use a window mean on every second of data? I'm working on a dataset right now and these differences are showing up in the final results.

Line 162: "An own"? Perhaps replace with "We developed a method,.." or something of the sort.

Line 174-175: Incomplete sentence "…the perturbed and template images is achieved."

Figure 3: Does the colorbar in (d) go with (c)? Or are these colors just representing shape? It seems like the structure annotated in (d) has a mean negative temperature, but based in the colorbar in (c) this structure would have a mean positive temperature. Maybe just make a note in the caption that says (c) only represents the shape of the structure.

Line 244: "1-m"

Line 278: Comma "…, as well as"

Line 292: put the equation in paratheses (=\overline{w'T'})

Line 316: Comma "complex,"

Line 341: "The information provided in Fig. 5 is visualized spatially in Fig. 6." Isn't figure 5 the spatial standard deviation of T' written in line 329, and figure 6 is the temporal standard deviation of each pixel?

Section 3.4: This is my favorite section. Good work!

Line 430: Just make sure you're consistent with putting your variables in italics

Line 472: "14-m" Use a dash when using numbers as an adjective.

Line 523: Recording was at 30 Hz? Or are you saying the inertial subrange is captured with 10 Hz measurements?

---

## Author Response (AR2)

**Dear Reviewer,**

It was likewise a pleasure for the authors of this work to receive Your welcoming reviews and encouraging comments. We wish that the positive but careful approach exercised by You were used more often in the community!

My responses are given in blue.

Line 32: "sizes" to "size". - done

Line 33: "durations" to "duration". - done

Line: 34 remove "now". - done

Line 37: remove "already" - done

Line 46: "depend" to "depends". - done

Line 100: "of" to "is a" - the DJI camera XT2 is du al-sensor, i.e. it combines both a thermal and an RGB sensor in one body. That is why I wrote "thermal sensor of XT2 is FLIR Tau 2".

Line 123: What do you mean by "decimated"? Did you cherry pick every 30th image, or did you use a window mean on every second of data? I'm working on a dataset right now and these differences are showing up in the final results. - The original 30 Hz thermal sequences were downsampled to 1 Hz, i.e. every 30th image was kept. Averaging over the 30 images recorded each second is unnecessary in my opinion, as there is no noise that needs elimination on that level - and even harmful, as small-scale structures would become lost. I hope to upgrade to the 10 Hz data analysis in a future study, but this will require a reduction of errors emerging in image co-registration. The upper limit on meaningful frequencies is, of course, imposed also by the random thermal noise of the camera, and may be lower than 10 Hz.

Line 162: "An own"? Perhaps replace with "We developed a method,.." or something of the sort. - done

Line 174-175: Incomplete sentence "...the perturbed and template images is achieved." - corrected; the sentence reads now "The process is continued until convergence between the perturbed and template images is achieved";

Figure 3: Does the colorbar in (d) go with (c)? Or are these colors just representing shape? It seems like the structure annotated in (d) has a mean negative temperature, but based in the colorbar in (c) this structure would have a mean positive temperature. Maybe just make a note in the caption that says (c) only represents the shape of the structure. – added to the figure 3 caption: The colors in (c) are only to tell the structures apart and do not correspond to the color bars in the other panels. Line 244: "1-m" - done

Line 278: Comma "..., as well as" - done

Line 292: put the equation in paratheses  $(=\langle w'T' \rangle)$  - done

Line 316: Comma "complex," - done

Line 341: "The information provided in Fig. 5 is visualized spatially in Fig. 6." Isn't figure 5 the spatial standard deviation of T' written in line 329, and figure 6 is the temporal standard deviation of each pixel? –Actually, this is the same quantity, T', presented in Fig. 5 as a statistical distribution, and in Fig. 6 spatially by plotting the maps of its per-pixel standard deviation. Section 3.4: This is my favorite section. Good work! - Thanks a lot for the good words! Line 430: Just make sure you're consistent with putting your variables in italics - done Line 472: "14-m" Use a dash when using numbers as an adjective. - done

Line 523: Recording was at 30 Hz? Or are you saying the inertial subrange is captured with 10 Hz measurements? I would say that the inertial subrange in the atmospheric boundary layer starts at frequencies much smaller than 1 Hz (actually at around 0.01 Hz), 10 Hz is already well into the dissipation subrange. The camera does record at 30 Hz, but this is certainly an excessive rate for the atmospheric applications.

Kind regards,

Pavel Alekseychik